EMBO
Molecular Medicine

# Enriched environment enhances β-adrenergic signaling to prevent microglia inflammation by amyloid-β

Huixin Xu[1] , Molly M Rajsombath[1], Pia Weikop[2] & Dennis J Selkoe[1,*]

## Abstract

**Environmental enrichment (EE) is a rodent behavioral paradigm that can model the cognitive benefits to humans associated with intellectual activity and exercise. We recently discovered EE's anti-inflammatory protection of brain microglia against soluble oligomers of human amyloid β-protein (oAβ). Mechanistically, we report that the key factor in microglial protection by EE is chronically enhanced β-adrenergic signaling. Quantifying microglial morphology and inflammatory RNA profiles revealed that mice in standard housing (SH) fed the β-adrenergic agonist isoproterenol experienced similar protection of microglia against oAβ-induced inflammation as did mice in EE. Conversely, mice in EE fed the β-adrenergic antagonist propranolol lost microglial protection against oAβ. Mice lacking β1/β2-adrenergic receptors showed no protection of microglia by EE. In SH mice, quantification of norepinephrine in hippocampus and interstitial fluid showed that oAβ disrupted norepinephrine homeostasis, and microglial-specific analysis of β2-adrenergic receptors indicated a decreased receptor level. Both features were rescued by EE. Thus, enhanced β-adrenergic signaling at the ligand and receptor levels mediates potent benefits of EE on microglial inflammation induced by human Aβ oligomers *in vivo*.**

**Keywords** Alzheimer's disease; environmental enrichment; microglia; neuroinflammation; β-adrenergic signaling

## Introduction

The steady aging of the human population and a resultant rise in the prevalence of Alzheimer's disease (AD) have made AD among the most common and feared diagnoses in medicine. Pharmaceutical approaches have had very limited success to date, and even when a safe and effective disease-modifying agent is approved, the cost of its chronic administration and its complex delivery logistics

may still leave many elderly humans underserved. Population studies increasingly suggest that lifelong experiences such as education, challenging occupation, exercise, and socialization may all improve cognitive reserve and provide some protection against the development of AD later in life (Nithianantharajah & Hannan, 2009; Then *et al*, 2015).

Environmental enrichment (EE) is an animal housing paradigm aimed at understanding the biological mechanisms that build cognitive reserve (Nithianantharajah & Hannan, 2006; Pang & Hannan, 2013; Hannan, 2014), thereby providing insights for developing non-pharmaceutical therapeutic approaches to AD. In rodents, EE has been shown to ameliorate multiple AD-like neuronal phenotypes, including synaptic loss, impaired synaptic plasticity, decreased neurogenesis, and altered cognition (Lazarov *et al*, 2005; Cracchiolo *et al*, 2007; Herring *et al*, 2009; Li *et al*, 2013). Environmental enrichment provides resistance to neuronal/synaptic toxicity from exposure to soluble oligomers of amyloid β-protein (oAβ), the most genetically and biologically well-validated pathogenic factor in AD (Selkoe & Hardy, 2016).

A major feature of AD pathobiology that has barely been studied with regard to EE is the brain's innate immune system. Changes in microglia contribute importantly to the AD phenotype both in human studies and in animal models (Zhang *et al*, 2013). Genome-wide association studies (GWAS) had identified over 20 AD-associated gene variants by 2016, and a majority were expressed by microglia (Villegas-Llerena *et al*, 2016). We recently showed for the first time that EE confers extensive anti-inflammatory effects on microglia in mice examining very early events in oAβ toxicity (Xu *et al*, 2016). Elucidating the molecular mechanisms by which EE dampens oAβ-induced microglial inflammation could not only support behavioral approaches but also identify key signaling paths that might provide new pharmacological targets to slow or prevent AD.

Norepinephrine (NE) and β-adrenergic signaling are centrally involved in neuronal activity in cognition and, more recently, in microglial immune function. NE is released by noradrenergic neurons that innervate many brain regions, including the dentate gyrus (DG) where we observed the most robust influences of EE on microglia (Xu *et al*, 2016). Although no direct evidence has been

1   Ann Romney Center for Neurologic Diseases, Brigham and Women's Hospital & Harvard Medical School, Boston, MA, USA
2   Center for Translational Neuromedicine, University of Copenhagen, Copenhagen, Denmark
    *Corresponding author. Tel: +1 617 525 5200; E-mail: dselkoe@bwh.harvard.edu

reported to connect EE with NE in the DG of healthy mice, the NE system has been found to regulate various features of cognitive activity (Sara, 2015) that are strongly enhanced by EE. A previous study from our laboratory found that EE activates β2-adrenergic receptors (β2-AR) in neuronal synapses as an early part of its neuroprotective effect against oAβ, suggesting links that β2-AR signaling has with both EE and oAβ toxicity (Li *et al*, 2013). Microglia express β1- and β2-adrenergic receptors (Mori *et al*, 2002; Tanaka *et al*, 2002), and their activation by exogenous agonists causes changes in microglia morphology and suppression of the pro-inflammatory cascade induced by lipopolysaccharide stimulation (Hetier *et al*, 1991; Dello Russo *et al*, 2004; Markus *et al*, 2010; Qian *et al*, 2011). Furthermore, NE was shown to suppress microglial inflammation and increase phagocytosis of synthetic Aβ42 fibrils in cell culture, while NE deficiency in APP transgenic mice enhanced brain tissue cytokine levels and exacerbated AD-like cytopathology (Heneka *et al*, 2002, 2006, 2010; Kalinin *et al*, 2007; Kong *et al*, 2010).

Here, we report multiple lines of evidence that β-adrenergic signaling drives the anti-inflammatory benefits of EE on microglia early in the *in vivo* response to oAβ isolated directly from human (AD) brain. By combining selective housing with neuropharmacological treatments, we first show that a β-adrenergic receptor agonist mimics EE's anti-inflammatory effects in wild-type mice housed in standard cages (SH) and exposed to intracerebroventricular (i.c.v) microinjections of human oAβ. Conversely, an antagonist to β-adrenergic receptors largely blocks such benefits of EE. Mice with germline knockouts of both β1- and β2-AR also lost EE's anti-inflammatory protection of microglia, in accord with the mice fed propranolol. Further, we find that, in SH mice, oAβ significantly increases norepinephrine level in the dentate gyrus and decreases it in brain interstitial fluid, and it downregulates microglial β1/β2AR levels. We conclude that environmental enrichment upregulates hippocampal β-adrenergic signaling to provide robust protection of microglia against the inflammatory effects of human oAβ oligomers.

# Results

### Prolonged oral administration of isoproterenol to SH mice prevents human oAβ-induced microglia inflammation *in vivo*, mimicking the protection by EE

Microglia express significant levels of β1/β2AR, suggesting these receptors have functional roles in microglial biology. To examine the mechanism underlying EE's strongly protective effect against the *in vivo* microglial reaction to oAβ that we recently reported (Xu *et al*, 2016), namely EE significantly rescues microglial morphological changes and inflammatory RNA profile shifts induced by exposure to oAβ, we asked whether prolonged activation of β1/β2AR signaling could mimic the immunomodulatory effects of EE on microglia. To achieve constant *in vivo* stimulation of β1/β2AR without causing stress to the animals, we administered isoproterenol, a non-selective β1/β2AR agonist, to 4-weeks SH mice continuously in their daily drinking water (0.1 g/l) for 4–6 weeks, the same period we have used for EE exposure. The compound is tasteless, and nontransparent water bottles were used for isoproterenol solution and plain water. No difference in water consumption was observed

between SH mice on isoproterenol and on plain water, although the exact amount of water consumed by individual mouse was untraceable which may contribute to some variations among mice of the same treatment group. Also, no significant differences in activity level and body weight were observed.

We first analyzed microglia morphology in SH mice fed isoproterenol or not and then microinjected i.c.v with oAβ-rich soluble cortical extracts (ADTBS) prepared from clinically and neuropathologically typical AD patients (Shankar *et al*, 2008). TBS extracts prepared identically from age-matched normal human cortex lacking oAβ (Fig EV1) while containing similar basal levels of cytokines (Appendix Fig S1) served as the control injectates (designated Control-TBS). After 48 h (Xu *et al*, 2016), we immunostained contralateral hemisphere cryosections for P2ry12 and CD68 as microglial markers. The contralateral hemisphere was exposed to equal amounts of oAβ as the ipsilateral hemisphere by the i.c.v. injections (Appendix Fig S2) and was therefore studied to avoid any confounding tissue injury effects from the ipsilateral injection site. As described previously (Xu *et al*, 2016), we performed Z-stack confocal microscopy in the dentate gyrus (DG), the region with the most robust EE benefits and strongest morphological responses toward oAβ shown by our previous work, and compressed the stacking images into one 2D image for comprehensive visualization of all microglial processes. We quantified microglia for their cell density (#microglia/mm$^2$), circularity, solidity, %CD68 reactivity per microglia (defined as the percentile of P2ry12$^+$ pixels of each microglia that are also CD68$^+$), and #branches per microglia using ImageJ for unbiased morphometry. In the untreated SH mice exposed to oAβ *in vivo*, we observed significantly decreased microglial density (vs. Control-TBS injections), increased microglial circularity and solidity, decreased #branches/microglia, and increased %CD68/microglia, all of which indicate a more inflammatory microglial state (Kettenmann *et al*, 2011). These results are entirely consistent with our recent report of i.c.v oAβ effects in SH mice (Xu *et al*, 2016). In contrast, SH mice fed with isoproterenol for 4–6 weeks showed significantly lower inflammatory changes in their microglial morphology on all the aforementioned measurements, as shown both by the original confocal images and Imaris 3D-reconstructed individual microglia cells (Fig 1A and C). Indeed, the microglial morphometry of the isoproterenol-fed SH mice exposed to oAβ yielded similar results to those of plain water-fed EE mice exposed to oAβ (Xu *et al*, 2016), indicating that isoproterenol treatment has microglial immunosuppressive capacity toward oAβ similar to that of EE. Isoproterenol-fed vs. untreated SH mice showed no significant differences in microglial morphology after injection of Control-TBS extracts (Fig 1B and C). Thus, the ADTBS-induced differences in microglial morphology described above are associated with oAβ. Images with all channels displayed individually are included in Appendix Fig S3.

Next, we performed mRNA expression profiling by NanoString on microglia isolated by gradient fractionation and fluorescent-activated cell sorting (FACS) after the i.c.v. injection of ADTBS. To provide a baseline for normalization and cross-comparison with minimal risk of identifying genes responding to patient-to-patient differences in the brain extract instead of to oAβ, an aliquot of the same patient's ADTBS extract immunodepleted (ID) of its Aβ by the 4G8 monoclonal antibody was i.c.v injected into littermate mice within each study. An Aβ$_{x-42}$ ELISA developed in our laboratory

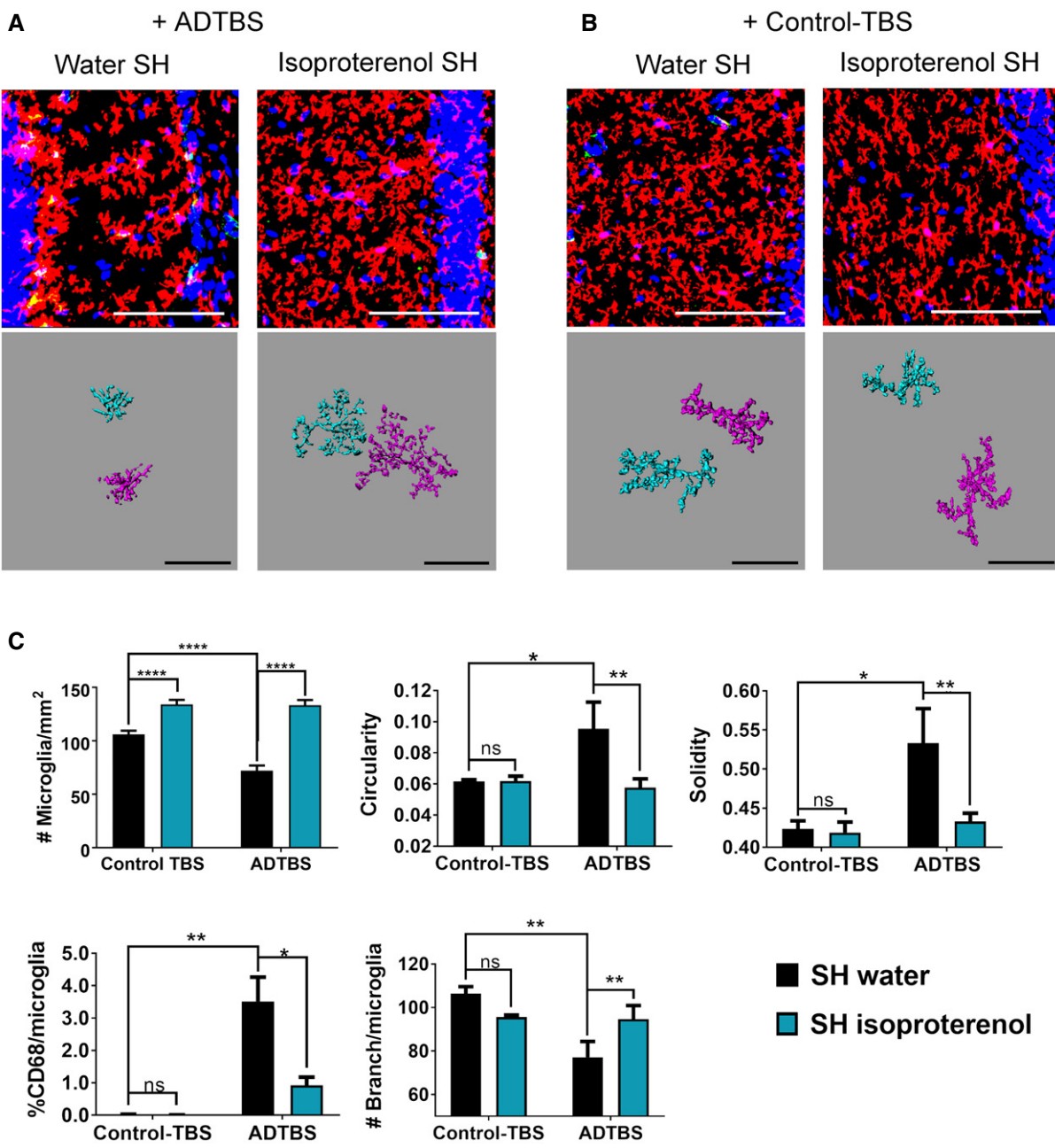

**Figure 1.  Isoproterenol treatment successfully neutralizes oAβ-induced microglia morphology change in SH mice.**

A, B    Representative images from SH mice treated with isoproterenol vs. water and injected with ADTBS and control-TBS. When injected with ADTBS, SH mice treated with isoproterenol have microglia with significantly less inflammatory morphological features comparing to their water control. When injected with control-TBS, SH mice under both treatments showed minimal microglia inflammatory features. Red: P2ry12; green: CD68; blue: DAPI. Scale bar = 100 μm. Corresponding Imaris 3D-reconstructed images featuring two cells per condition are presented under each panel. The spatial relation between each two cells may be adjusted to fit in the frame with maximum magnification. Scale bar = 50 μm.

C    Quantification analysis of SH microglia morphology under different treatments. In control SH mice, ADTBS caused significant decrease in microglial density; increase in circularity, solidity, and %CD68; and decrease in branching complexity in comparison with control-TBS (*$P < 0.05$; **$P < 0.01$; ****$P < 0.0001$, $N = 6$). Isoproterenol treatment significantly rescued all above-mentioned morphological features (*$P < 0.05$; **$P < 0.01$; ****$P < 0.0001$, $N = 6$). All quantitative data are presented as mean $\pm$ SD. Exact $P$ values are listed in Appendix Table S1. All statistical analysis were performed using multiple $t$-test (unpaired, do not assume equal SDs) with Holm-Sidak method to determine significance.

(Yang *et al*, 2013) indicated that ID-ADTBS had < 4% of the total Aβ42 in the original ADTBS (Fig EV1) while having no significant differences in endogenous human cytokines (Appendix Fig S2),

indicating the inflammatory differences we observed are Aβ-induced. The microglia from a 3 mm cubic block of the contralateral hemisphere directly symmetrical to the ipsilateral injection site were

FACS-isolated 48 h after the i.c.v injections using $CD11b^+/CD45^{Med}/Ly6C^{Lo}/Ly6G^{Lo}$ as sorting markers, and their RNA samples were analyzed by NanoString nCounter Mouse-inflammatory (V2) CodeSet containing 249 inflammatory genes and six housekeeping genes. Only genes with higher-than-background expression were included in further analysis. Each individual microglial sample from the ADTBS-injected group was normalized to the average of the whole control-injected group (ID-ADTBS) for each of the two treatments (SH water and SH isoproterenol). Ratios were presented in the heat map after conversion to Log2 format. Positive values (red) indicate a relative increase in an mRNA due to oAβ exposure, while negative ones (blue) indicate relative suppression by oAβ. We found that untreated SH mice had significantly altered inflammatory gene mRNA profiles upon oAβ exposure, while in striking contrast, isoproterenol-fed SH mice had a largely unaltered and even slightly immunosuppressive mRNA profile (Fig 2A), similar to what we have observed in EE mice maintained on regular drinking water (Xu *et al*, 2016). Twenty-nine genes that were significantly altered by oAβ exposure in water-fed SH mice were significantly neutralized by the isoproterenol feeding (analyzed using the ADTBS/ID-ADTBS ratio; results *after* Holm–Sidak correction; Fig 2B). An additional 19 genes showed significant neutralization by isoproterenol only before the Holm–Sidak correction (Fig 2C). Raw NanoString values prior to their normalization are found in the Dataset EV1. No significant differences were observed between isoproterenol-treated and control SH mice that had no oAβ injection, which is consistent with our previous observations in the EE vs. SH paradigm (Xu *et al*, 2016).

In summary, our results so far demonstrate that prolonged oral exposure to isoproterenol in SH mice can successfully protect against human oAβ-induced microglial inflammation *in vivo*. The microglial responses to oAβ in isoproterenol-treated SH mice are more "EE-like", strongly suggesting that norepinephrine signaling helps mediate EE's anti-inflammatory benefit for microglia.

**Prolonged oral exposure of EE mice to propranolol significantly diminishes the protective benefits of EE on microglia**

To assess further whether β1/β2AR signaling is required for EE's anti-inflammatory benefits, we fed EE mice for 4–6 weeks with propranolol (0.4 g/l in the drinking water), a non-selective βAR antagonist shown to block EE's benefits for neuronal function in previous work on oAβ (Li *et al*, 2013). No sign of decreased physical activity, body weight, and food/water intake was observed in the propranolol-treated EE mice. Similar to the case of isoproterenol treatment, the exact amount of water consumed by individual mouse was untraceable which may contribute to some variations among mice of the same treatment group.

Using the same analytical system as described in the prior section, we first analyzed microglial morphology from the EE mice fed propranolol or not and then injected i.c.v. with oAβ-rich AD-TBS. As expected, water-fed EE mice showed prevention of oAβ-induced effects on microglial morphology in the contralateral hippocampus DG area, as shown both in the original confocal images and in Imaris 3D-reconstructed microglia cells (Fig 3A and C), as evidenced by minimal changes in cell density (#microglia/mm²), circularity, solidity, and branching complexity, as well as a modest %CD68 increase. In contrast, propranolol-fed EE mice showed significant morphological changes upon oAβ exposure that

were similar to those of SH mice on all of the aforementioned measures of individual microglial morphology, with decreased microglial density, increased microglial circularity and solidity, markedly decreased #branches/microglia, and much more increased %CD68 than untreated EE mice, again as shown both by the original confocal images and Imaris 3D-reconstructed microglia cells (Fig 3A and C). These data show that propranolol-fed EE mice have microglial morphologies corresponding to a more inflammatory state, i.e., a more "SH-like" state, upon oAβ injection than do water-fed EE mice. Again, propranolol-treated vs. untreated EE mice showed no difference in microglia morphology upon injection of Control-TBS extract (Fig 3B and C), suggesting that the ADTBS-induced differences in morphology represent a response to oAβ. Images with all channels displayed individually are included in Appendix Fig S4.

Next, mRNA profiling of FACS-purified microglia after ADTBS vs. oAβ immunodepleted (ID)-TBS injection was performed on the NanoString nCounter Mouse-inflammatory (V2) CodeSet. The resultant heat map (Fig 4A) includes all genes with higher-than-background expression and was generated following the same rules as those in Fig 2A (red indicates relative increase in expression by oAβ over the ID control, while blue indicates relative suppression). Side-by-side comparison of the propranolol-fed and untreated EE mice revealed significantly altered inflammatory mRNA profiles upon oAβ exposure by propranolol (Fig 4A). Indeed, 13 genes whose expression levels were successfully neutralized by EE upon oAβ exposure were now significantly stimulated or suppressed in the propranolol-treated EE mice (analyzed using ADTBS/ID-ADTBS ratio; results *after* Holm–Sidak correction; Fig 4B), suggesting a more pro-inflammatory expression state. In addition, 19 other genes also fit into this category before but not after Holm–Sidak correction (Fig 4C). Raw NanoString values prior to normalization are included in the Dataset EV2. Like what we observed in isoproterenol- vs. water-treated SH mice and in SH vs. EE mice, propranolol alone did not induce any significant microglial inflammatory gene profile shift.

In summary, propranolol significantly blocked EE's immunomodulatory benefits to microglia upon oAβ exposure, rendering the EE microglia more "SH-like". Collectively, our data strongly suggest a quantitatively important and indispensable involvement of noradrenergic signaling in mediating the EE-microglia effects.

In the two NanoString profile studies in SH isoproterenol vs. water paradigm and EE propranolol vs. water paradigm, we highlighted Ccl2, Ccl3, Ccl4, Tnf, and Cxcl10 because they were among the most significantly altered genes under *both* paradigms and are well studied for their biological functions in inflammation, making them reliable and robust representatives to evaluate microglial inflammatory status in this and future studies. To confirm their increase in response to oAβ stimulation at the protein level, we quantified the levels of CCL2, CCL3, CCL4, and CXCL10 in ADTBS- vs. ID-ADTBS-injected brain tissues by ELISA and observed significant increases in all four cytokines upon ADTBS exposure (Fig EV2). TNFα was not detectable at the protein level with our 1 pg/ml detection limit.

We then attempted to confirm our findings by using a more purified oAβ preparation from human brain that lacks small molecules (e.g., glutamate and drugs), the brain levels of which could vary from patient to patient. For this confirmatory step, we adopted a

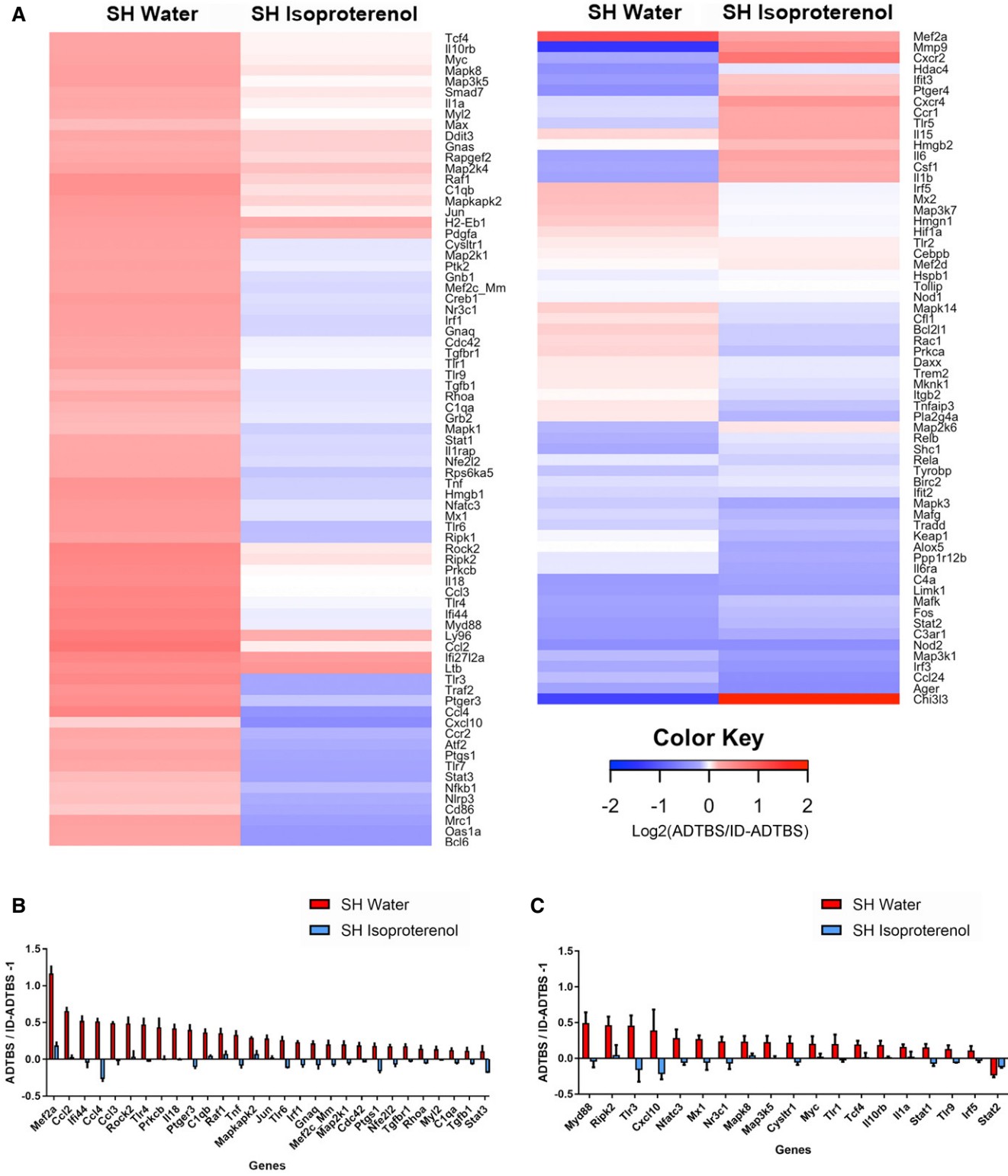

**Figure 2. Isoproterenol treatment significantly rescues oAβ-induced microglial inflammatory gene profile shift in SH mice.**

A    Heat map of all microglial inflammatory genes with expression level above background cutoff by NanoString nCounter analysis. All data are presented by Log₂ (ADTBS/ID ratio).

B, C    Forty-seven genes are significantly rescued by isoproterenol treatment among 140 genes actively expressed by microglia. Twenty-nine of them are significant ($P < 0.01$) with Holm–Sidak correction for multi-comparison (B), and 19 of them are significant ($P < 0.01$) without correction (C). $N = 6$. All data are presented as mean ± SD. Exact $P$ values are listed in Appendix Table S2.

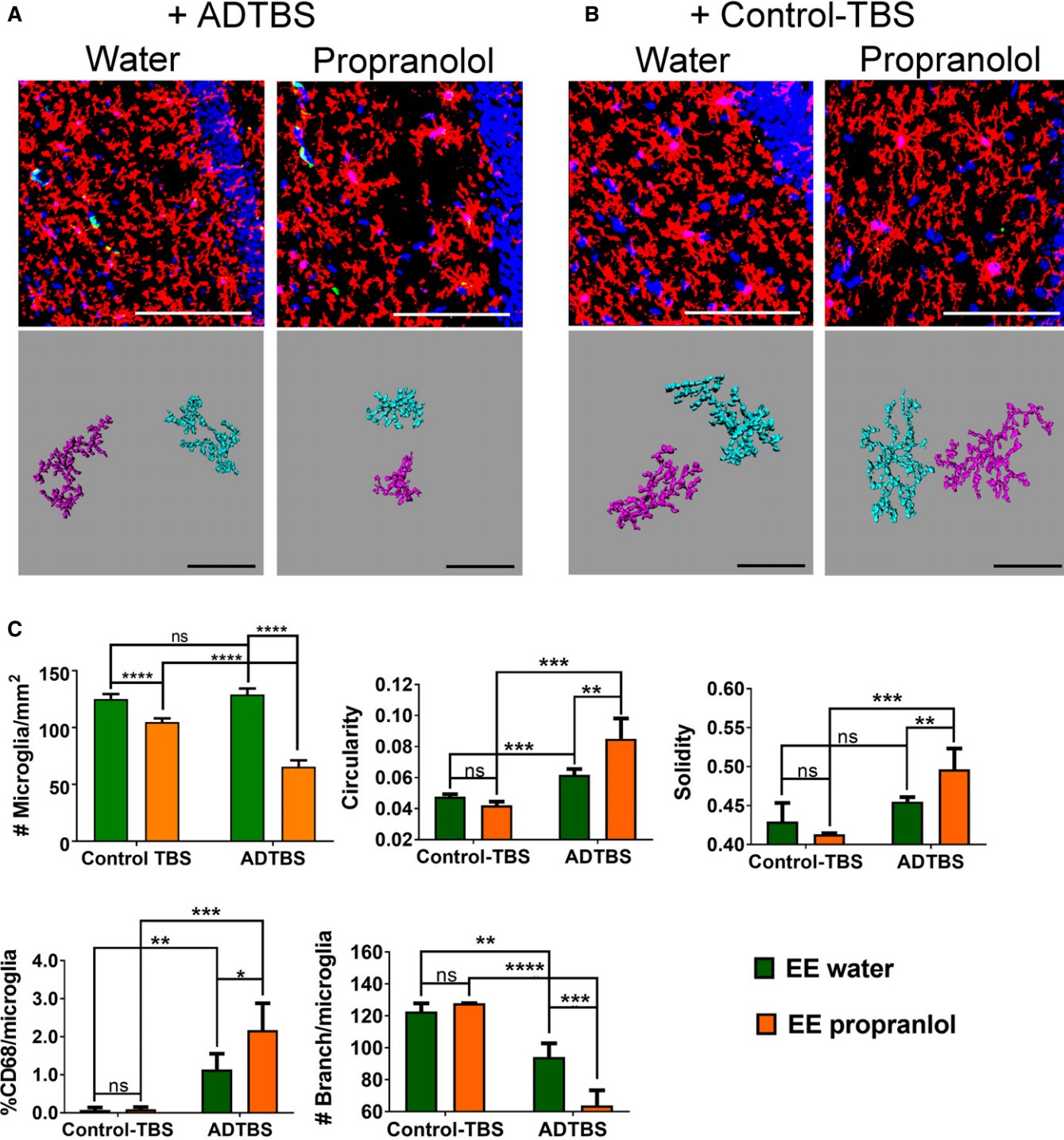

**Figure 3. Propranolol treatment significantly diminishes EE's immunosuppressive effects on microglia against oAβ-induced morphology change.**

A, B  Representative images from EE mice treated with propranolol vs. water and injected with TBS extract from ADTBS and control-TBS. When injected with ADTBS, EE mice treated with propranolol have microglia with significant inflammatory morphological features, while their water controls showed much more physiological microglia. When injected with control-TBS, EE mice under both treatments showed minimal microglia inflammatory features. Red: P2ry12; green: CD68; blue: DAPI. Scale bar = 100 μm. Corresponding Imaris 3D-reconstructed images featuring two cells per condition are presented under each panel. The spatial relation between each two cells may be adjusted to fit in the frame with maximum magnification. Scale bar = 50 μm.

C  Quantification analysis of EE microglia morphology under different treatments. Propranolol-treated EE mice had significantly decreased microglial density, increased microglia circularity, solidity, and %CD68, while decreased branching complexity after oAβ injection comparing to their water-treated controls (*$P < 0.05$; **$P < 0.01$; ***$P < 0.001$; **** $P < 0.0001$, N = 6). All quantitative data are presented as mean ± SD. Exact *P* values are listed in Appendix Table S3. All statistical analysis were performed using multiple *t*-test (unpaired, do not assume equal SDs) with Holm-Sidak method to determine significance.

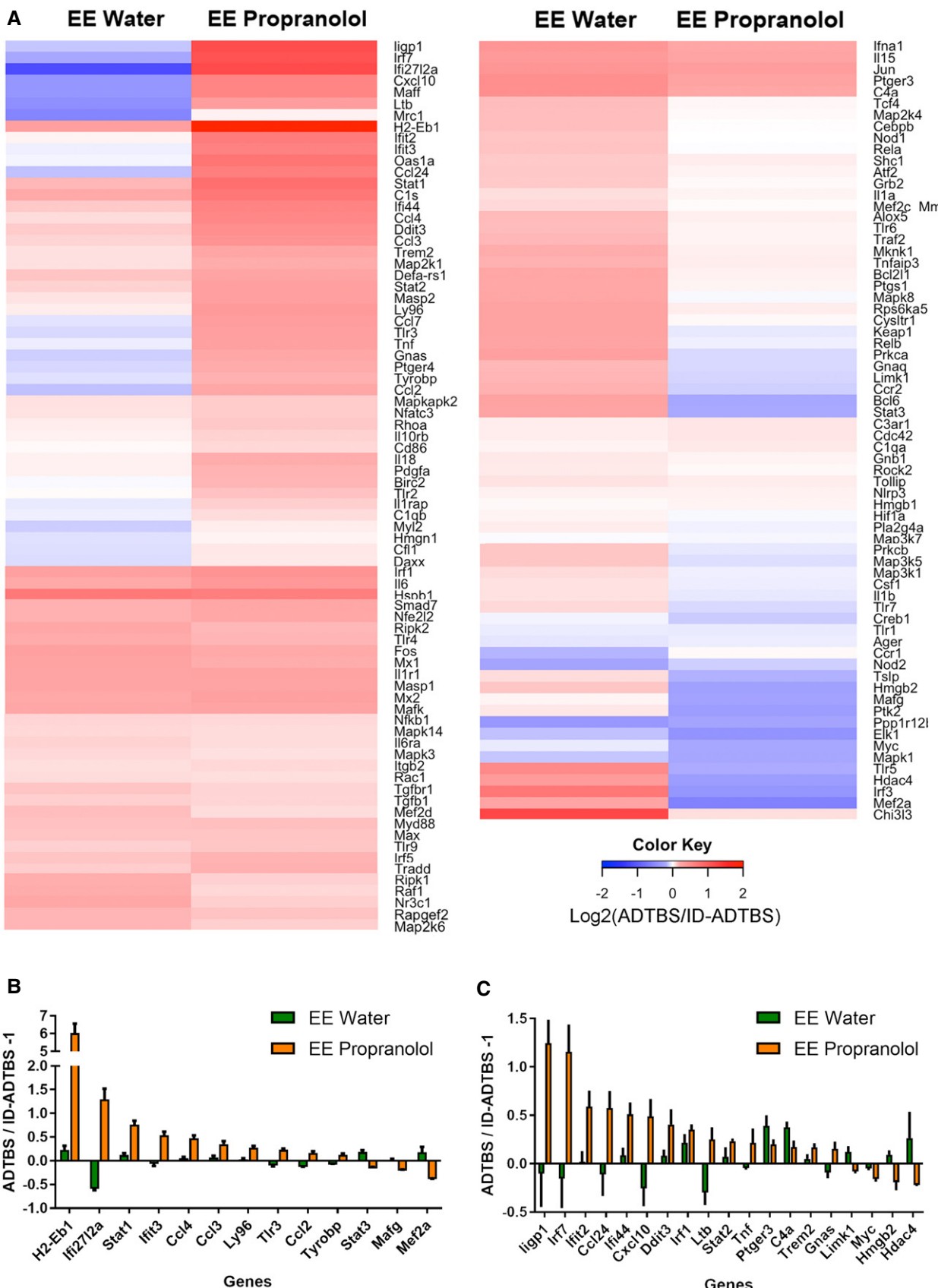

Figure 4.

◄    **Figure 4.    Propranolol treatment significantly diminishes EE's immunosuppressive effects on microglia against oAβ-induced microglial inflammatory gene profile shift.**

A    Heat map of all microglial inflammatory genes with expression level above background cutoff by NanoString nCounter analysis. All data are presented by $Log_2$ (ADTBS/ID ratio).

B, C    Thirty-two genes are significantly altered by propranolol treatment among 160 genes actively expressed by microglia. Thirteen of them are significant ($P < 0.01$) with Holm–Sidak correction for multi-comparison (B), and 19 of them are significant ($P < 0.01$) without correction (C). $N = 6$. All data are presented as mean $\pm$ SD. Exact $P$ values are listed in Appendix Table S4.

previously published oAβ purification protocol that relies on using size-exclusion chromatography (SEC) to enrich oAβ in the "void volume" fraction during a quantitative removal of small molecules, followed by 37°C incubation of that fraction to dissociate high MW Aβ oligomers to lower MW oAβ comprising dimers and moderately larger oligomers, which we have found to have particularly high bioactivity on microglia and neurons *in vivo* (Yang *et al*, 2017). To ensure the injected material had a physiological background, all incubated oAβ-rich solutions were lyophilized and reconstituted in sterile phosphate-buffered saline (PBS), pH 7.4, before *in vivo* injections. The reconstituted samples were quantified using an $Aβ_{x-42}$ ELISA. Subsequent adjustment of the reconstitution volume was done so that each mouse received ~4 pg of human $Aβ_{x-42}$ *in vivo*, a dose we found to induce significant increases of key cytokines in SH mice by 24 h after i.c.v. injection. Selective immunodepletion of Aβ successfully removed the bioactivity of this void volume-derived oAβ, rendering it indistinguishable from blank vehicle (PBS; Fig EV3).

Using this SEC-isolated, low MW Aβ oligomer preparation and its vehicle serving as the negative control, we observed microglial activation patterns consistent with those obtained from the *in vivo* studies of oral isoproterenol or propranolol treatment in SH mice or EE mice using the whole brain TBS extracts. Specifically, transcript levels of the five cytokines highlighted by NanoString in both our prior treatment paradigms (above) were quantified by qPCR. Isoproterenol-treated SH mice showed significant prevention of microglial inflammation upon this SEC-purified oAβ injection, whereas propranolol-treated EE mice lost their protection (Fig 5A and B). Note that with this purified material, we observed much more robust activation of all five genes than with the unfractionated ADTBS, possibly due to the enrichment and the reduced influence of other brain-derived molecules. Further NanoString analysis on those samples confirmed that the SEC-purified oAβ induces consistent gene profile patterns like ADTBS vs. ID-ADTBS (Fig EV4).

### β1/β2AR germline knockout mice fail to gain EE-mediated protection of microglia against oAβ

To confirm the apparently critical role of functional noradrenergic receptor signaling in mediating the EE effects against microglial oAβ toxicity, we acquired from Jackson Laboratory mice with a germline global β1/β2AR deletion (homozygous null for both Adrb1 and Adrb2 genes, referred to here as Adrb1/2KO). The mice were bred in-house and exposed to EE starting at 4 weeks of age for a total of 4–6 weeks, identical to the EE paradigm we conducted above using wild-type mice. At the end of EE, the mice were given ADTBS or control-TBS by i.c.v. microinjection. Brains were harvested and analyzed for microglial morphology as before.

We observed statistically significant signs of inflammation (decreased microglial density, increased circularity/solidity, decreased

branching complexity, and increased $CD68^+$ lysosomal alteration) upon ADTBS exposure vs. control-TBS in the DG of Adrb1/2KO SH mice. In stark contrast to our earlier findings in wild-type mice, no quantifiable difference between SH and EE environments was observed in Adrb1/2KO mice for all aforementioned morphological parameters upon *in vivo* oAβ exposure, as shown both by the original confocal images and by Imaris 3D-reconstructed microglia cells, indicating a complete loss of EE's protection against microglial inflammation (Fig 6A and B). Images with all channels displayed individually are included in Appendix Fig S4. Similar morphological changes were also observed in CA of hippocampus (Fig EV5), but without obvious changes in microglial density, which is also consistent with that of WT SH mice as reported in our previous work (Xu *et al*, 2016).

Next, we performed i.c.v. injections of the SEC-purified human brain low MW oAβ fractions (vs. PBS) into EE vs. SH Adrb1/2KO mice and analyzed the expression levels of the top five cytokines (Ccl2, Ccl3, Ccl4, Cxcl10, and Tnf) by qPCR. oAβ now produced indistinguishable pro-inflammatory responses in both SH and EE KO mice (Fig 6C). Collectively, the morphometry and expression analyses indicate that the lifelong absence of β1/β2AR leads to almost complete loss of the beneficial modulation of microglial phenotype by EE, consistent with our earlier observations in EE mice chronically treated with propranolol.

### Quantification of norepinephrine in mice reveals disrupted NE signaling in SH but not EE mice

We quantified norepinephrine (NE) in the DG of wild-type mice kept in SH or EE. Knowledge of NE properties in the mouse dentate gyrus during EE is lacking. We first measured NE levels in dissected whole DG tissue. The DG was rapidly dissected on ice and snap-frozen until analysis by high-performance liquid chromatography with electrochemical detection (HPLC-ECD). The results showed that when the mice were injected i.c.v. with the SEC-purified oAβ (vs. PBS), a small (~20%) but highly significant increase in DG NE level of SH mice was observed at 24 h, whereas no detectable change in NE occurred in EE mice (Fig 7A).

Next, we sampled interstitial fluid (ISF) from the DG by *in vivo* microdialysis to specifically quantify extracellular NE levels. The mice underwent implantation of a microdialysis guide cannula into the hippocampal dentate gyrus contralateral to the side of the i.c.v. injection of purified human oAβ or PBS. ISF samples were collected 24 h after injection at 1-h intervals using a push-only system with a 1-mm-long, 6 kDa molecular weight cutoff (MWCO) dialysis membrane and 1 μl/min perfusion of artificial CSF (Fig 7B). ISF samples were collected at 4°C and then snap-frozen until later analysis by HPLC-ECD. The room was kept quiet with minimal activities during ISF sampling to avoid confounding effects on norepinephrine production. To minimize any technical differences during microdialysis, each oAβ-injected mouse was paired with a PBS-injected

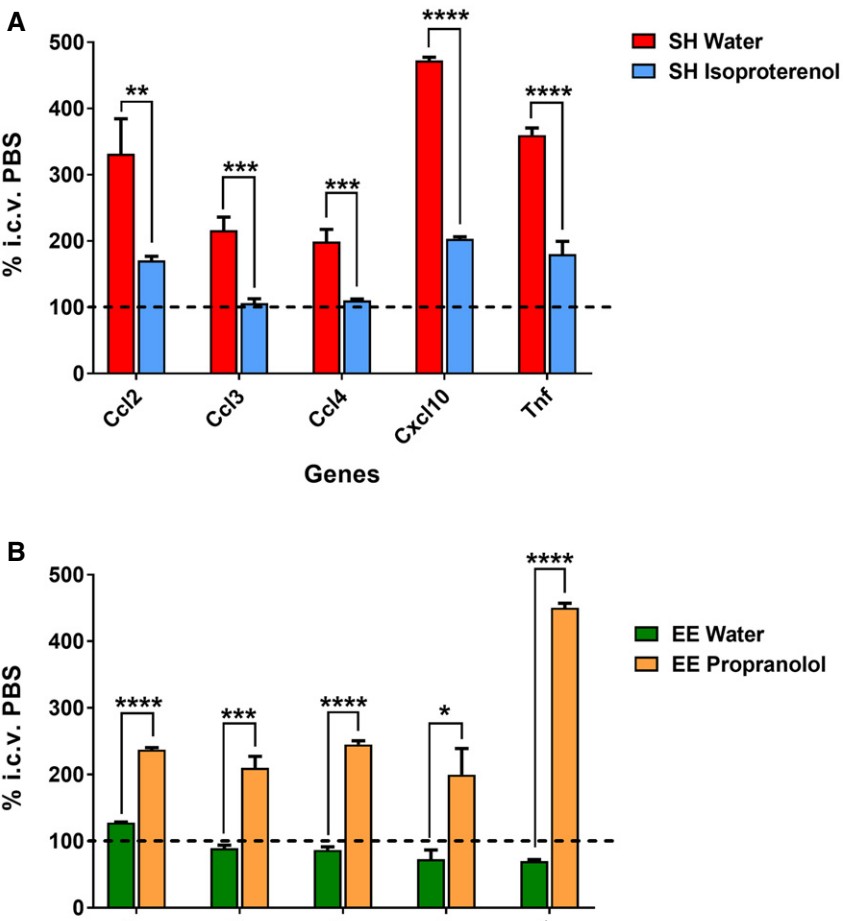

**Figure 5. SEC-purified AD brain extract can recapitulate the phenotypes observed using straight AD brain extract in SH and EE mice orally treated with isoproterenol or propranolol.**

A  In SH mice, five top cytokines highlighted by NanoString screening are significantly increased by injection of SEC separated oAβ vs. PBS, measured by qPCR. Isoproterenol treatment successfully neutralizes such inflammatory responses.

B  In EE mice treated with propranolol, the same five top cytokines are significantly increased by injection of SEC separated oAβ vs. PBS, measured by qPCR. Control EE mice showed complete rescue consistent with previous studies.

Data information: All data are presented as mean ± SD. *$P < 0.01$; **$P < 0.001$; ***$P < 0.0001$; ****$P < 0.00001$. $N = 6$. Exact $P$ values are listed in Appendix Table S5. All statistical analysis were performed using multiple $t$-test (unpaired, do not assume equal SDs) with Holm–Sidak method to determine significance.

mouse from the same housing environment (SH or EE), and both these mice were attached simultaneously to a single shared micro-dialysis perfusion pump. All oAβ-injected mice that had been living in either SH or EE housing had their NE levels normalized to the average NE level of all PBS-injected mice from that same housing state, thereby generating a percentage. Mice with any bleeding noticeable at the microdialysis surgical site prior to probe insertion were excluded. Strikingly, we found that, in SH mice, i.c.v. human oAβ induced a significant ~50% decrease in extracellular (ISF) NE level, but in EE mice, no effect of oAβ was observed (Fig 7C).

Furthermore, qPCR analysis of the post-i.c.v. injection contralateral microglia indicates that both β1-AR and β2-AR transcription levels decreased ~50% in SH mice after oAβ injection but did not change at all in EE mice (Fig 7D). We were able to identify one β2AR antibody of a specific lot that successfully detects microglial β2AR despite the very low protein yields. When pooling microglia harvested

from four contralateral brain blocks as described above, we were able to show that oAβ injection caused a complete loss of the β2AR band by Western blot in SH mice but not in EE mice (Fig 7E; raw blot images in Appendix Fig S6), confirming our findings at the RNA level. No β1AR antibody was found to perform under these conditions, but because of the similarities between β1AR and β2AR, we speculate that β1AR should follow the same type of change as β2AR.

Collectively, these data point to a substantially disrupted NE signaling system in SH mice exposed to human oAβ, as shown by increased total tissue [NE] but lowered free extracellular [NE] and decreased availability of β1/β2AR in the dentate gyrus. In striking contrast, EE mice maintained stable NE levels in both whole tissue and ISF and unchanged β1- and β2-AR transcript levels upon oAβ injection, indicating that under environmental enrichment, unaltered noradrenergic homeostasis is associated with the prevention of microglial inflammation.

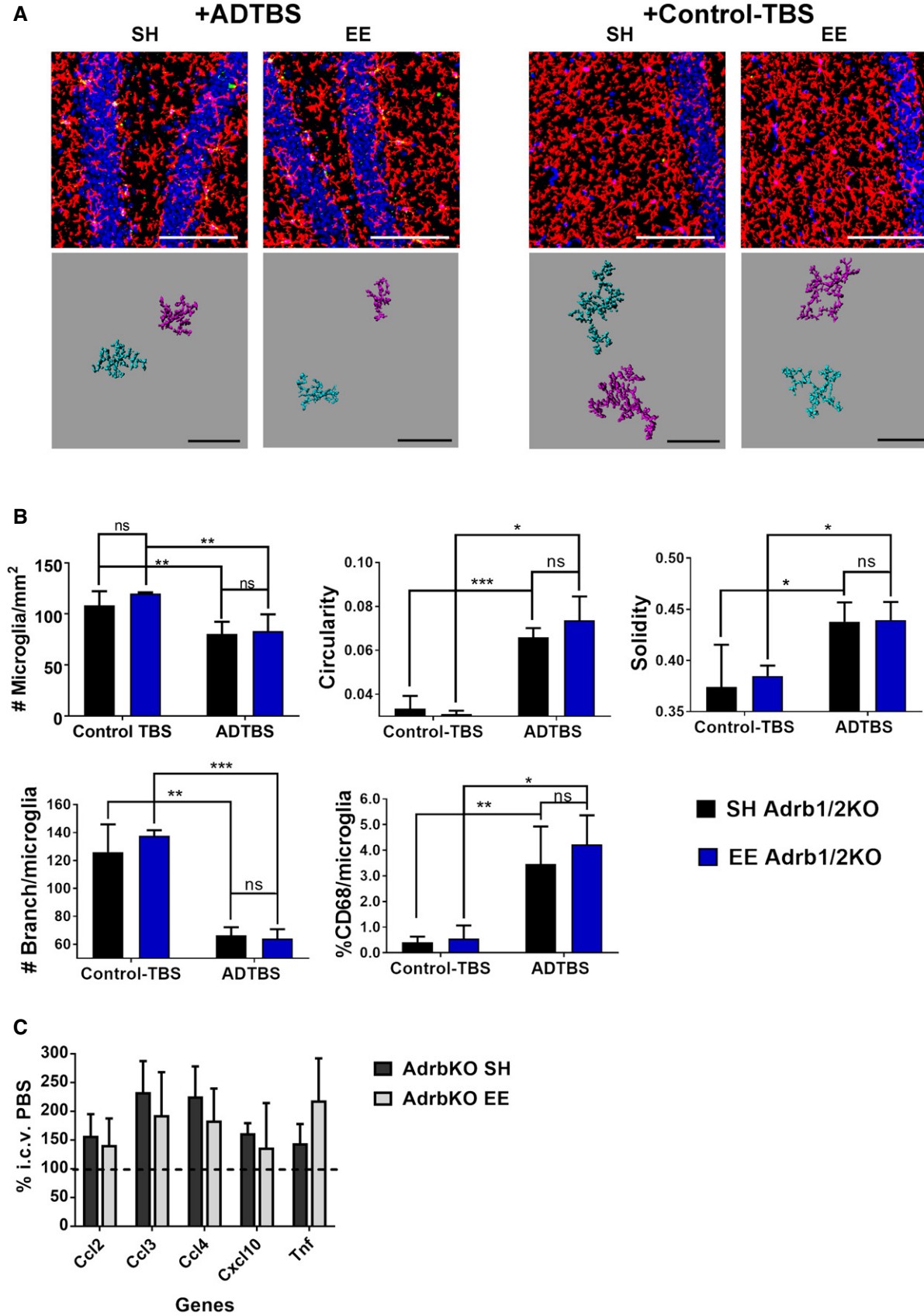

Figure 6.

**Figure 6.  Germline knockout of β1 and β2AR (Adrb1/2 KO) leads to loss of EE protection on microglia with acute oAβ exposure.**

A    Representative images from SH and EE Adrb1/2KO mice injected with ADTBS and control-TBS. When injected with ADTBS, both SH and EE mice have microglia with significant inflammatory morphological features in DG and CA. When injected with control-TBS, both SH and EE mice showed minimal microglia inflammatory features. Red: P2ry12; green: CD68; blue: DAPI. Scale bar = 100 μm. Corresponding Imaris 3D-reconstructed images featuring two cells per condition are presented under each panel. The spatial relation between each two cells may be adjusted to fit in the frame with maximum magnification. Scale bar = 50 μm.

B    Quantification analysis of microglia morphology in DG. Both SH and EE Adrb1/2KO mice had significantly decreased microglial density, increased microglia circularity, solidity, and %CD68, while decreased branching complexity after oAβ injection comparing to their controls (*$P < 0.05$; **$P < 0.01$; ***$P < 0.001$, $N = 5$). Exact $P$ values are listed in Appendix Table S6.

C    qPCR analysis of SH and EE Adrb1/2KO mice microglia after i.c.v. injection of SEC separated oAβ vs. PBS. Both SH and EE mice showed increase in cytokine expression with oAβ injection. $N = 4$. No significant difference between SH and EE is observed.

Data information: All quantitative data are presented as mean ± SD. All statistical analysis were performed using multiple *t*-test (unpaired, do not assume equal SDs) with Holm-Sidak method to determine significance.

Lastly, we tested whether the cAMP/PKA pathway, a common signaling pathway utilized by β1/β2AR, mediates the downstream effects. We inhibited cAMP and PKA pathways by one cAMP inhibitor (cAMP-Rp, 50 μM) and two PKA inhibitors (KT5720, 100 nM; PKA inhibitor fragment (6-22) amide (marked as PI (6-22)), 100 nM) in primary microglia cultures treated with oAβ and isoproterenol (100 μM). All doses were chosen to be within fivefold of the compound's IC$_{50}$ without any cytotoxicity. The cells were harvested after 4 h. Because our culturing system (20,000 cells/well in 150 μl culture media) does not yield sufficient amount of cytokines at protein level to be detected by ELISA, we analyzed the treated primary microglia by qPCR for the expression of the five cytokines highlighted by our previous NanoString analysis (Ccl2, Ccl3, Ccl4, Cxcl10, and Tnf). We found that 10 pg of oAβ induced significant increases of all five cytokines by 4 h in the primary cultures, and isoproterenol had significant rescue effects. cAMP-Rp significantly diminished the anti-inflammatory effects of isoproterenol in Tnf expression but not as much for the other four cytokines. Both PKA inhibitors failed to reverse any isoproterenol effects (Fig 7F). The findings suggest that the downstream signaling of β1/β2AR's anti-inflammatory effects on oAβ-stimulated microglia is only partially cAMP-dependent and not PKA-independent.

## Discussion

Mounting evidence in humans suggests potential cognitive benefits detectable in old age of intellectually challenging and physically active lifestyles. With the widespread availability of disease-modifying drugs for Alzheimer's disease still years away, understanding the biological mechanisms underlying lifestyle benefits could lead to valuable additional approaches to lessen the global crisis of age-related dementia and could simultaneously enable discoveries of new pharmacological targets other than the key pathological proteins currently being targeted in AD clinical trials. Environmental enrichment is a laboratory adaptation to model effects of human lifestyle. It has been studied in rodent models for its potential benefit in ameliorating neuropathological features resembling those of AD. We recently documented potent immunomodulatory effects of EE in protecting microglia from the early phase of oAβ-induced inflammation *in vivo*, thereby expanding EE's known benefits from purely neuronal to the realm of innate immunity (Xu *et al*, 2016). One other study offered support for EE-microglia effects by reporting enhanced microglial clearance of Aβ plaques by EE in 5 × FAD mice, revealing another feature of EE's modulation of microglia when the cells are chronically exposed to excessive levels of oAβ (Ziegler-Waldkirch *et al*, 2018).

Pro-inflammatory microglial activities are believed to produce various detrimental consequences in the brain and help promote neurodegeneration. One very recent study demonstrated that inflammatory microglia promote Aβ aggregation *in vivo* by releasing ASC (apoptosis-associated speck-like protein containing CARD) specks into extracellular space, offering yet another strong piece of evidence for the destructive power of microglial inflammation (Venegas *et al*, 2017). The recent identification of microglial gene variants contributing to late-onset AD in a GWAS study of 85,000+ subjects further highlighted the importance of microglia in AD pathogenesis (Sims *et al*, 2017). The molecular mechanisms driving EE's immunomodulation of microglia, once identified, could inform the development of synergistic behavioral and pharmacological approaches to downregulate microglial inflammation and neutralize its damage, but, until the current study, the responsible biological mechanism remained known.

Here, we have identified β-adrenergic signaling as the apparent key mediator of EE's highly consistent benefit of preventing oAβ-induced microglial inflammation in the living brain. Our findings from a combined comparison of housing paradigms and pharmacological treatments indicate that prolonged daily exposure to isoproterenol successfully prevented SH mice from developing microglial inflammation upon exposure of oAβ, while prolonged exposure of EE mice to propranolol significantly diminished EE's protective effects and allowed much greater microglial inflammation from oAβ even though the EE mice treated with propranolol still maintained their daily activity and had a body habitus indistinguishable from their untreated littermates. These data clearly demonstrate that a very substantial part of EE's immunosuppression on microglia is driven via β1/β2AR signaling. Blocking β1/β2AR signaling prevented acquisition of microglial immuno-protection. None of our drug treatments or the EE paradigm alone induced any significant microglial inflammatory gene profile changes when oAβ was not present, indicating that the regulation of microglia via β1/β2AR signaling is more relevant to microglial inflammatory status than to physiological status. Importantly, these pharmacological results were confirmed through genetic manipulation when we observed a consistent loss of EE benefits on oAβ-exposed microglia in mice completely lacking functional β1/β2ARs. While a microglia-specific Adrb KO mouse line is not yet available, the recently identified microglia-specific marker, Sall1, and the successful downregulation of other microglial genes via Sall1-Cre (Buttgereit *et al*, 2016) indicate that future research could potentially develop mice with selective deletion of microglial β-ARs in order to further dissect the functions of β-adrenergic signaling in the EE-mediated downregulation of microglial inflammation.

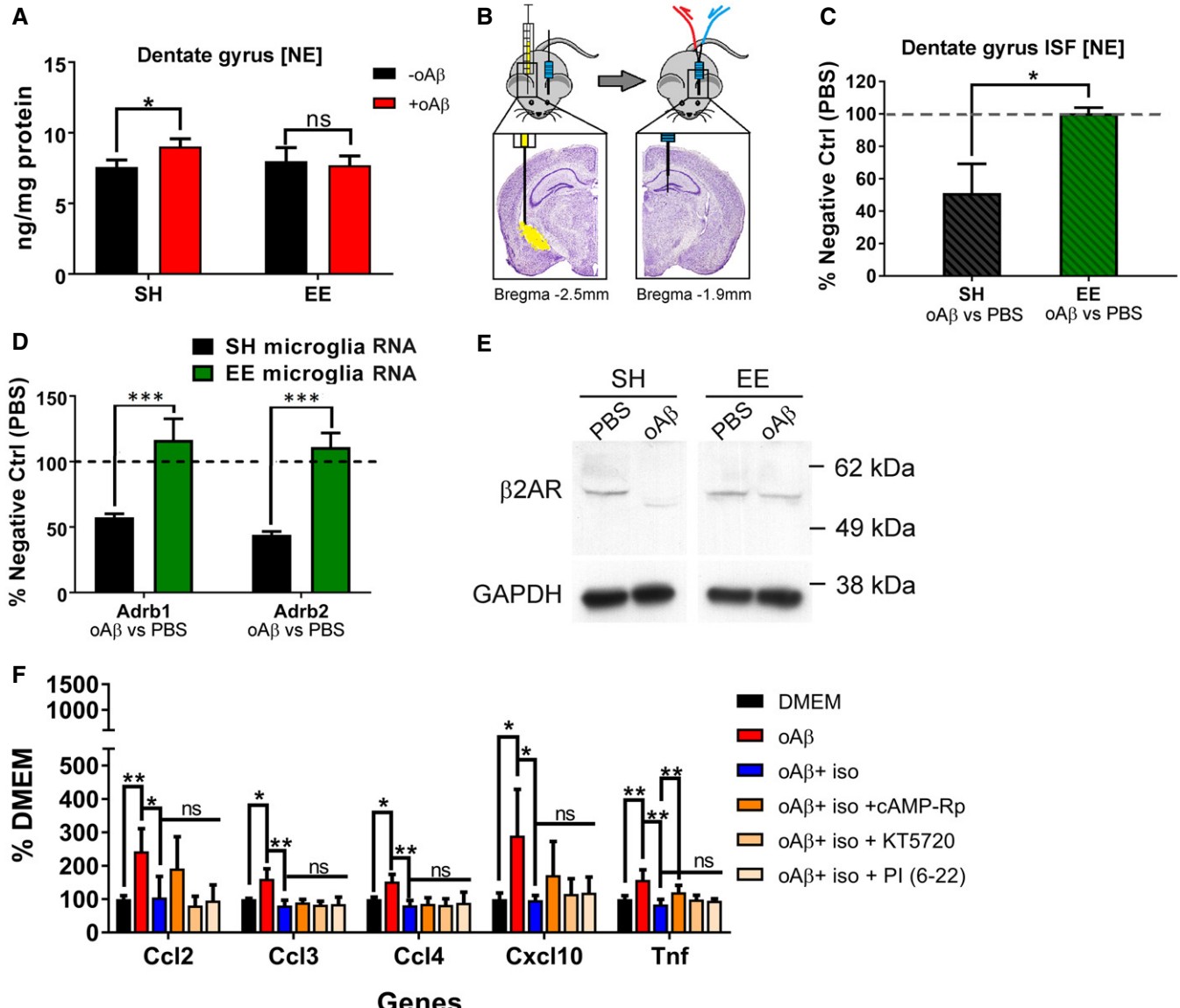

**Figure 7. Direct quantification of norepinephrine (NE) showed disrupted NE homeostasis by oAβ in SH mice but not in EE mice.**

A  Whole tissue NE quantification of the DG shows significant increase in NE level in SH mice injected with oAβ vs. PBS. EE mice showed minimal change in DG tissue NE level with oAβ injection vs. PBS. *P = 0.0083, N = 6.

B  Illustration for *in vivo* microdialysis from contralateral DG after i.c.v. injection. The brain images are adopted from Paxinos and Franklin (2012).

C  Quantification of NE in the ISF collected by *in vivo* microdialysis shows significant decrease in extracellular NE in the DG of SH mice with i.c.v. injection of oAβ vs. PBS, while EE mice showed minimal change. *P < 0.01 (P = 0.0060), N = 6.

D  qPCR analysis of microglial Adrb1 and Adrb2 RNA level showed significant decrease in both Adrb1 and Adrb2 transcription in SH mice after oAβ injection. EE fully rescued the phenotype. ***P < 0.0001 (Adrb1 P = 4.1E-05; Adrb2 P = 8.7E-07). N = 6.

E  Western blots of β2AR in SH and EE mice +/− oAβ injection. Each sample contained microglia isolated from four mice following the same protocol as that of NanoString analysis.

F  qPCR analysis of primary microglia culture treated with oAβ, isoproterenol, cAMP-Rp, KT5720, and PKA inhibitor fragment (6-22) amides (PI (6-22)) showed significant increase in cytokine expression by oAβ and robust rescue effects by isoproterenol. cAMP-Rp reversed isoproterenol protection in Tnf but not others. Both KT5720 and PKA inhibitor fragment (6-22) amides did not affect isoproterenol. *P < 0.05, **P < 0.01. N = 8. Exact P values are listed in Appendix Table S7.

Data information: All quantitative data are presented as mean ± SD. All statistical analysis were performed using multiple t-test (unpaired, do not assume equal SDs) with Holm-Sidak method to determine significance.

Next, we complemented these mechanistic findings by quantifying brain and ISF NE levels. Direct quantification of NE in whole DG tissue showed a small but significant increase in [NE] in SH mice upon exposure to oAβ. In contrast, microdialysis via a short probe embedded within the DG revealed that ISF NE levels, which represent the portion of total NE with impact on biological signaling, dropped by 50% in SH mice in response to oAβ. The increase in tissue NE and decrease in free extracellular NE in DG together point

to oAβ-impaired NE regulatory machinery in SH mice, a phenotype that was again successfully neutralized by EE. Besides reduced NE availability, we also observed strong decreases in the transcription levels of both adrenergic receptors and the protein level of β2AR from oAβ in SH mice, whereas minimal and insignificant changes occurred in EE mice. A decrease in the mRNA level of both these receptors has been reported in microglia stimulated with LPS (Gyoneva & Traynelis, 2013), connecting a pro-inflammatory state to the decrease in microglial β1/β2AR expression. Taken together, our data demonstrate for the first time that EE has a direct impact on the NE dynamics in hippocampal tissue and that EE suppresses excessive microglial inflammation in the early phase of oAβ exposure by maintaining the strength of β-adrenergic signaling in microglia. We speculate that oAβ toxicity may impair the NE release/ reuptake machinery in SH mice, leading to increased trapping of NE inside cells and a significant decrease in the adjacent extracellular space, and hence a loss of anti-inflammatory NE signal. The disturbance of NE homeostasis, together with a lowered β-AR level which exacerbates the loss of NE signal for microglia, renders the microenvironment in the SH hippocampus exposed to oAβ as pro-inflammatory. Whether the loss of β1/β2AR is a result of disrupted membrane receptor recycling could not be determined due to technical limitations in antibody sensitivity and protein amounts. However, our previous study in neurons successfully demonstrated increased β2AR internalization induced by oAβ (Li *et al*, 2013), suggesting it is possible that in microglia the dynamics of β1/β2AR follows the same pattern.

In this work, we solely used oAβ isolated directly from human (AD) brain tissue. Various sources of Aβ oligomers have been used in past studies, the vast majority of them being synthetic. Whether synthetic oAβ assemblies faithfully represent the pathological protein conformers accumulating in the human disease is major concern. Saline extracts of brain tissues from confirmed AD patients provide the most clinically relevant source of Aβ oligomers. Moreover, human oAβ is highly bioactive, perhaps ~100-fold more so than synthetic oAβ preps such as ADDLs when tested side-by-side on wild-type differentiated rat neurons (Jin *et al*, 2011). When administered i.c.v at picomolar doses to wild-type mice, oAβ-rich AD brain extracts provide the most disease-relevant model for the early oAβ toxicity encountered near the onset of the AD process. Using wild-type mice avoids confounding developmental issues from overexpression of pathogenic Aβ from embryonic stages on, as seen in APP transgenic mouse lines, and may serve as better models of the earliest features of sporadic human AD, providing more translational relevance to early disease mechanisms in humans than do APP-overexpressing transgenic models. Further, we adapted a previously published SEC purification protocol (Yang *et al*, 2017) to produce much purer oAβ solutions that were quantified, lyophilized, and reconstituted in a buffer of choice to the desired concentration for *in vivo* studies. Using this highly enriched and cleaner preparation, we could reproduce the same phenomena we had observed with cruder ADTBS extracts using different brains, with the quantitative differences among treatment groups accentuated. We believe this approach has facilitated consistent preparations and usage of endogenous human oAβ for future research here and elsewhere.

Propranolol has long been used in the clinic for medical conditions such as high blood pressure, certain arrhythmias, and even

stage fright. A few reports have suggested that β2AR activation can increase Aβ and tau pathology via regulating the internalization of surface APP molecules and/or γ-secretase activity, thereby increasing Aβ production, leading to a recommendation of using β-blockers such as propranolol in the clinical management of AD (Ni *et al*, 2006; Dobarro *et al*, 2013; Wang *et al*, 2013). However, more recent studies suggest that β-adrenergic signaling is instead protective (Heneka *et al*, 2010; Branca *et al*, 2014; Liu *et al*, 2015). The fact that locus coeruleus (LC) neurons develop tangles and die out in AD, leading to reduced NE signaling in humans (Matthews *et al*, 2002), is potentially relevant to our work. Indeed, lesioning LC in mice and rats increases Aβ cytopathology (Kalinin *et al*, 2007; Heneka *et al*, 2010; Jardanhazi-Kurutz *et al*, 2011). Moreover, one β1AR agonist was recently found to decrease overall brain inflammation and lessen AD pathology in an aggressive AD mouse model (Ardestani *et al*, 2017), providing strong support that β-adrenergic signaling plays a positive role in helping control AD-related brain injury. In addition to those studies utilizing exogenous agonist to the NE signaling, our discovery emphasizes that EE provides anti-inflammatory protection on microglia via endogenously enhanced β-adrenergic signaling independent of any pharmacological manipulation. The long established comprehensive benefits of EE to brain health, both in laboratory models and in human studies, strongly support the positive role of β-adrenergic signaling. In combination with earlier reports of decreased NE signaling in AD brain, our study highlights the potential importance of avoiding chronic beta-blockers in MCI and AD subjects. In this regard, a recent study conducted in > 10,000 nursing home denizens showed a strong correlation between usage of β-blockers and functional decline, and this negative effect was more obvious in residents with moderate or severe cognitive deficits (Steinman *et al*, 2017).

On the other hand, any potential benefit of long-term β-agonist usage (e.g., in treating asthma) for cognitive function in the elderly is not well documented. A recent epidemiology study identified β-agonist usage as a protective factor against Parkinson's disease (Mittal *et al*, 2017). Similar studies that focus on AD and dementia would be of great value, for although active lifestyle and constant intellectual challenge are highly recommended in the middle-aged and elderly populations to potentially lessen AD and dementia, in practice, such prolonged behavioral modification will not be accessible to all for physical and economic reasons. Our study identifies β-adrenergic signaling as the principal cellular event that is chronically enhanced by an enriched environment to prevent aberrant microglial inflammation occurring from oAβ in AD. Further clinical investigation and epidemiological studies of people who are prescribed β-agonists for chronic use could indicate whether such treatment leads to lower incidence and less rapid progression of mild cognitive impairment (MCI) or AD symptoms. Although a lifestyle modification approach has many superior aspects over pharmacological agents both physiologically and financially, such studies could be important for paving the road toward developing a safe β-agonist as a therapeutic for AD when lifestyle enrichment is not possible for certain patients.

In summary, our multifaceted findings reveal that β-adrenergic signaling is a major, if not dominant, mediator of the immunosuppressive effects of EE on oAβ-induced microglial inflammation. Moreover, we establish here a model for systematically investigating

signaling pathways and their functions on microglia in AD—using human brain-derived oAβ on non-genetically manipulated, wild-type mice. Our identification of NE signaling as a principal mediator between EE and microglia suggests that increased neuronal activity from EE stimulation may modulate microglial behavior through interactions of neurotransmitters with their cognate receptors expressed on microglia, providing evidence that CNS neurotransmitters can modulate not just neurons but also innate immune cells. NE signaling was previously identified as a major mediator of EE's protection of neuronal function following human oAβ exposure (Li *et al*, 2013). Our data suggest the existence of common cellular events triggered by EE that benefit both neurons and microglia via the same signaling pathway but with different manifestations. The beneficial role of enhanced β-adrenergic signaling identified here could apply to other neurological diseases, providing new insights into the biological underpinnings of EE and how to maximize its benefits for the brain.

# Materials and Methods

## Animals

The Harvard Medical School Standard Committee on Animals and BWH IACUC approved all experiments involving mice in this study. All mice were male. Animals were housed in a temperature-controlled room on a 12-h light/12-h dark cycle and had free access to food and water. BL6/129 mice were purchased from Taconic. Mice with germline Adrb1/2 knockout were purchased from Jackson Laboratory (Stock No: 003810) and bred in-house. All mice used in the experiments were 8- to 10-weeks old.

## Antibodies

The following antibodies were used for this study: P2ry12 (1:500), a generous gift from Dr. Oleg Butovsky; CD68 (1:200), Abcam ab53444, RRID:AB_869007; 4G8 (20 μg/reaction), Biolegend® RRID: AB_10175149; Anti-CD11b (1:100), BD Biosciences Cat# 552850 RRID:AB_394491; anti-CD45 (1:200), eBioscience Cat# 17-0451-82 162 RRID:AB_469392; anti-Ly6C (1:400), eBioscience Cat# 12-5932-82 RRID:AB_10804510; anti-Ly6G (1:100), Biolegend 163 Cat# 127605 RRID: AB_1236488; GAPDH (1:5,000), Sigma-Aldrich G9545 RRID: AB 796208; β2AR (1:500), Abcam ab182136 (lot: GR302897-20) RRID: AB not available.

Detailed methods regarding environmental enrichment setup, mouse brain tissue staining and microglial morphology analysis, microglia isolation and RNA analysis, and human brain extract preparation and analysis are reported by previous publications (Yang *et al*, 2013, 2017; Xu *et al*, 2016). Methods unique to this study are listed below:

## Oral administration of propranolol and isoproterenol

Mice were orally exposed to isoproterenol or propranolol through the daily drinking water for 4–6 weeks. Isoproterenol was administered at 0.1 g/l, and propranolol was administered at 0.4 g/l. Colored, light-proof water bottles were used in all groups. Total water consumption of each cage was recorded. Note that the water

consumption of individual animals cannot be determined and therefore may add variation to the study.

## Mouse cytokine quantification at protein level

Mice were injected i.c.v. with ADTBS or ID-ADTBS. The brain tissue was harvested following the same method as that for Nano-String analysis. The brain blocks were homogenized in TBS with protease inhibitor at 1:2 ratio (1 mg wet tissue: 2 ml homogenization buffer). The brain TBS homogenates were centrifuged at 50,000 *g* for 1 h at 4 degree to remove all non-water-soluble fractions. The supernatant fractions were analyzed by mouse cytokine ELISA kit from Meso Scale Discovery (K15069L). For each sample, the loading volume was adjusted based on total protein concentration determined by BCA protein assay so each ELISA reaction received same amount of total protein.

## Human cytokines quantification

All human brain extracts were analyzed by LEGNEDplex™ Human Inflammation Panel to cover major cytokines that may be present in human brains.

## Stereotactic intracerebroventricular (i.c.v.) injection

The lateral ventricle was located by stereotactic coordinates of bregma−2.5 mm, midline+3.1 mm, and dura−3.7 mm in wild-type mice, and bregma−2.4 mm, midline+3.0 mm, and dura−2.7 mm in Adrb1/2 KO mice. Through a Hamilton syringe (25 μl) in a stereotactic holder, 4 μl of injectate was slowly delivered over 5 min. When this was combined with microdialysis, the ventricle coordinates were changed to bregma−2.5 mm, midline−3.1 mm, and dura−3.7 mm in wild-type mice to accommodate the microdialysis probe placement.

## Microscopy and image analysis

All images were acquired by Zeiss LSM710 confocal microscope using 20× objective. A Z-stack was collected for each field of view to capture all details in microglial processes. Microglia morphological features were quantified by ImageJ particle analysis and Skeleton analysis. Imaris reconstruction was performed using Imaris software with the same Z-stacks.

## *In vivo* microdialysis

These mice had an intracerebral guide cannula implanted using coordinates for right hippocampal dentate gyrus placement (bregma−1.9 mm, midline−1.1 mm, and dura−1.5 mm). After inserting probes with 6 kDa molecular weight cutoff (MWCO) membranes (CMA), artificial CSF (in mM: 1.3 CaCl₂, 1.2 MgSO₄, 3 KCl, 0.4 KH₂PO₄, 25 NaHCO₃, and 122 NaCl, pH 7.35, filtered and degassed) was perfused at flow rates 1.0 μl/min with an infusion syringe pump (Stoelting). Microdialysates were collected in a fraction collector at 4°C every hour. Mice were housed in a Raturn cage system (Bioanalytical Systems), which allowed normal movements and activity. All ISF samples were snap-frozen and stored at −80°C until further analysis.

**Quantification of norepinephrine**

All mouse brain samples were snap-frozen upon dissection and analyzed at the Neurochemistry Core Facility at Vanderbilt University by Dr. Ginger Milne. All interstitial fluid (ISF) samples from *in vivo* microdialysis were analyzed at the Center for Translational Neuromedicine (University of Copenhagen, Denmark) by Dr. Pia Weikop. All the ISF samples were processed on Prodigy 3 μm ODS-3 C18 column (2 mm × 100 mm, particle size 3 μm, Phenomenex). The mobile phase consisted of 55 mM sodium acetate, 1 mM octanesulfonic acid, 0.1 mM Na2EDTA, and 8% acetonitrile, adjusted to pH 3.2 with 0.1 M acetic acid, and was degassed using an online degasser. Seven μl of the samples was injected, and the flow rate was 0.15 ml/min. Electrochemical detection was accomplished using an amperometric detector (Antec Decade, Antec, Leiden, NL) with a glassy carbon electrode set at 0.8 V, with a Ag/AgCl electrode as reference electrode. The output was recorded on a computer program LC Solution from Shimadzu, which was also used to calculate the peak areas.

**Western blot of microglial β2AR**

Brain microglia were purified as described by Xu *et al* (2016). Microglia from four mice were pooled and lysed in 40 μl of RIPA lysis buffer and treated for 30 min at 37°C with 2 units of DNase (TURBO, Invitrogen AM2239) to remove DNA released during lysing. The samples were then mixed with 4X SDS and incubated at room temperature for 10 min and boiled at 65°C for 5 min. The Western blots were developed using Pierce™ Fast Western Blot Kits, SuperSignal™ West Femto. GAPDH was used as loading control.

**Preparation of primary microglia and treatment**

Primary microglia cultures were prepared from P0 mouse pups following a published protocol (Lian *et al*, 2016). Briefly, the cortices of the brains were harvested and digested by trypsin. The cells were plated in poly-D-lysine (PDL)-coated T-75 flasks as mixed glial cultures and let grow till confluent astrocyte layers form. The microglia were isolated by tapping the flasks vigorously. Purified microglia were seeded into PDL-coated 96-well plates at 20,000 cells/well density and let recover overnight. The cells were treated with 10 pg of SEC-purified oAβ, 100 μM isoproterenol, 50 μM cAMP-Rp (Tocris 1337), 100 nM KT5720 (Tocris 1288), and 100 nM PKA inhibitor fragment (6-22) amide (Tocris 1904). Cells were harvested after 4 h of incubation at 37°C and lysed for RNA analysis.

**Statistical analysis and heat maps**

All quantification results were analyzed using multiple *t*-test (unpaired, do not assume equal SDs). Statistical significance was determined using the Holm–Sidak method, with alpha = 0.05. All data were analyzed by Shapiro–Wilk test first to determine normality (all passed). N indicates total number of mice under each condition. The sample sizes were determined by power analysis based on pre-specified effect and variation estimated from preliminary studies. Each experiment included animals from two or more independent

**The paper explained**

**Problem**
Alzheimer's disease (AD) is affecting a rapidly growing population globally. A pharmacological solution still faces multiple challenges, but lifelong experiences such as education, challenging occupation, exercise, and socialization are shown to provide protection against the development of AD later in life. Environmental enrichment (EE), a laboratory model to study such lifestyle benefits, has extensive anti-inflammatory effects on brain microglia whose malfunction under amyloid-β (Aβ) challenge is tightly connected to the progression of AD both in animal models and human subjects. Identifying the molecular mechanisms that drive EE's protective effects on microglia against Aβ is critical for offering new insights into therapeutic development and gaining knowledge on the communication and regulation between the environment and brain innate immune system to benefit future research on brain health.

**Results**
Here, we provide evidence that enhanced β-adrenergic signaling is the key behind EE's anti-inflammatory benefits on microglia under Aβ assault. We show that oral treatment of β-adrenergic receptor (β-AR) agonist mimics EE's benefits on microglia in mice housed in standard housing (SH), while oral treatment of β-AR antagonist to environmental enriched mice diminished EE's protection. Genetically removing β-AR also blocks the mice from gaining microglial benefits from EE. Further, we demonstrate that SH mice experienced disrupted β-adrenergic signaling with Aβ exposure shown by decreased extracellular norepinephrine and decreased level of microglial β-AR. EE maintains the β-adrenergic signaling strength in mice with Aβ exposure, therefore keeping brain microglia at a less inflammatory state.

**Impact**
The identification of β-adrenergic pathway activation as one major molecular event mediating EE's effects on microglia connects outside environment to brain innate immune system by neurotransmitter signaling. The finding might open up new research into the neuro-microglia communication via neurotransmitters and have clinical implications regarding usage of β-agonists and β-antagonists to treat other diseases and their potential cognitive impact in the population vulnerable to AD. Further research might decide whether the same mechanism can benefit other neurological disorders where aberrantly activated microglia play pivotal roles in disease progression.

cohorts to eliminate cohort-specific bias. The initial sample sizes were larger than calculated to compensate for any possible exclusion of data points, which was minimal. Any animal with significant bleeding during/after surgery was excluded from the study to avoid confounding from excessive tissue injury. For all experiments, the animals were assigned to different groups by random selection and the animal caregiver was not the investigator. Sample processing and data analysis were carried out by investigators blinded to the intervention. Heat maps were generated using the heatmap.2 function in the gplots package in R.

**Expanded View** for this article is available online.

## Acknowledgments

We thank Dr. Ginger Milne at Vanderbilt University Neurochemistry Core for performing analysis on tissue NE level. We thank Dr. Oleg Butovsky for providing P2ry12 antibody. The study was supported by NIH grant AG006173 (to DJS).

## Author contributions

HX and DJS designed the experiment. HX, MMR, and PW conducted the experiments. HX and DJS interpreted the data and prepared the manuscript.

## Conflict of interest

The authors declare that they have no conflict of interest.

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
