## [Review Process File · EMBO Molecular Medicine]

Enriched environment enhances β -adrenergic signaling to prevent microglia inflammation by amyloid- β

Huixin Xu, Molly M. Rajsombath, Pia Weikop, Dennis J. Selkoe

Review timeline:

Submission date:	29 January 2018
Editorial Decision:	08 March 2018
Revision received:	07 June 2018
Editorial Decision:	03 July 2018
Revision received:	12 July 2018
Accepted:	18 July 2018

Editor: Céline Carret

Transaction Report:

1st Editorial Decision

08 March 2018

Thank you for the submission of your manuscript to EMBO Molecular Medicine. We have now heard back from the three referees whom we asked to evaluate your manuscript.

You will see that while the referees find the study to be important for the field, they also highlight issues and potentials concerns. Referee 1 is not convinced by the experimental design and requests more data to back up the conclusions drawn. Referee 2 suggests discussing the limitations of the study and providing better imaging. Referee 3 has more substantial concerns and would like to see more mechanism to increase the novelty of the findings.

I therefore would welcome the submission of a revised version within three months for further consideration and would like to encourage you to address all the criticisms raised as suggested to improve conclusiveness and clarity, experimentally when requested. Please note that EMBO Molecular Medicine strongly supports a single round of revision and that, as acceptance or rejection of the manuscript will depend on another round of review, your responses should be as complete as possible.

I look forward to receiving your revised manuscript.

***** Reviewer's comments *****

Referee #1 (Remarks for Author):

Xu et. al. investigated the role of β -adrenergic signalling via norepinephrine upon soluble A β -induced microglial activation in the context of environmental enrichment. In this study, the authors aimed to assess the molecular mechanism behind the benefits of physical and mental stimulation resulting in decreased risk of developing Alzheimer's disease (AD). This topic is, per se, of interest to the AD community as it might propose novel therapeutic avenues for the treatment of AD.

The authors, in line with previous work published by the group, argue that environmental enrichment (EE) of mice leads to reduced activation of microglia (assessed by microglial morphology and mRNA expression of pro-inflammatory genes) when exposed to soluble oligomeric A β species extracted from AD brains. The authors demonstrate that this protective effect can be replicated in standard housed (SH) mice by activating β -adrenergic receptors via isoproterenol (Figs. 1, 2). Similarly, blocking β -adrenergic receptors by propranolol administration in EE housed mice prevented the protective effect of EE (Figs. 3, 4). Next, mRNA expression of selected cytokines was replicated by administering a more purified patient-derived soluble oligomeric A β fraction (Fig. 5). To genetically confirm a role of noradrenergic receptors in this setting, the authors analysed β 1/ β 2-receptor knock out mice kept under SH or EE and treated with soluble A β ; here, the authors show that the beneficial effects of EE were not induced in these mice (Fig. 6). Lastly, the authors provide evidence that the beneficial effects of EE are mediated by the maintenance of norepinephrine signalling that is impaired in SH mice (Fig. 7). The authors thus claim that inducing β -adrenergic receptor activation pharmacologically or via physical or mental activity can prevent the damaging effects of microglia activation by A β oligomers.

As stated above the paper addresses an important topic in the field of AD research. However, there are some major concerns regarding the design of the study.

Major points

1. In order to draw conclusive data from the experiments, it would be required to have one single A β preparation paradigm and a consistent control for assessing both microglia morphology and gene expression:
 - a. In Fig. 1 and 3, microglial morphology is analysed subsequent to i.c.v. injection of soluble protein fractions from control and AD patients (Control-TBS vs AD-TBS).
 - b. In Fig. 2 and 4, microglial inflammatory gene expression profiles are assessed following i.c.v. injection of soluble protein fractions from AD patients with or without A β immunodepletion (AD-TBS vs ID-ADTBS).
 - c. In Fig. 5, microglial gene expression of selected inflammatory genes is analysed after i.c.v. injections of more purified A β preparations (PBS vs SEC-A β).

At least a thorough characterisation of the employed A β fractions is required. The following questions should be addressed:

- a. Is the amount of the soluble A β comparable to those used in the experiments in Fig.1 and 2 and in Fig. 3 and 4, respectively?
- b. Does the TBS fraction from the control patients also contain A β ?
- c. Are cytokines or cell debris present in the protein fractions?
- d. Why is the contralateral hemisphere used for analyses? Does the injected A β reach into the contralateral side? This needs to be confirmed experimentally prior to assessing those regions, which were used for the analyses of microglial morphology.

2. Figs. 2, 4, 5: The relevance of the NanoString data should be discussed in more detail.

- a. Why are the five selected genes verified by qPCR most representative for microglial activation and of utmost relevance for AD?
- b. Do the differentially expressed genes give any indication on downstream functional modifications?
- c. A NanoString analysis should be included for SEC-A β treated mice.
- d. Why are CD68 and P2ry12 not upregulated, which would be expected based on the IHC?
- e. Additional data on cytokine protein expression is required to support the notion that the treatment induces neuroinflammation, especially since the expression changes are pretty moderate.

3. Fig. 5 The authors claim in the text that they "were able to recapitulate the results obtained from

the *in vivo* studies of oral isoproterenol or propranolol treatment in SH mice or EE mice". The authors do not show any data supporting this claim; such data must be included.

4. Fig. 6 The authors argue for a specific effect of norepinephrine/noradrenergic receptor signalling on microglia, yet off-target effects are not assessed.
 - a. The authors refer to previous work demonstrating the expression of β -adrenergic receptors on microglia, however, these receptors are also expressed on astrocytes or nerve cells. These cells must therefore be analysed as well to see how they are affected by the different treatment paradigms.
 - b. A microglia-specific $\beta 1/\beta 2$ -receptor knock out would provide a better basis for the role of noradrenergic signalling in microglia and would prove that reduced microglia activation is not just a bystander of modulated noradrenergic signalling. Such - admittedly involved - experimental prove is required according to the opinion of this referee.
 - c. Microglia of the knockout mice seem to have no increased activation compared to the wild type microglia shown in Figs. 1 and 3. What is the author's explanation for this?

Minor points

1. Further editing/proofreading of the text should be undertaken to ensure complete understanding of the messages presented (e.g. first sentence of abstract).
2. Using an acute AD model the authors should discuss that reduced microglial activation translates to any AD-relevant functional or pathological changes.
3. Since the methods section mainly refers to previously published papers, it is necessary to provide more extensive methodological details in the text or figure legends, e.g.
 - a. Only in the paragraph describing Fig. 6 does it become clear how old the mice were when exposed to the varying housing conditions.
 - b. Which regions are stained in Fig. 1 and 3? Using the $\beta 1/\beta 2$ -knock out mice, in Fig. 6, the authors describe a more prominent effect on microglia morphology in the CA region compared to the DG. Why was this regional differentiation not made before?
 - c. What does the readout of %CD68/microglia indicate? CD68 is present in all microglia. Additionally, red, green and blue channels should be split in Fig. 1, 3 and 5 since a green CD68-positive signal is barely visible.
 - d. For NanoString analysis, which regions were the microglia isolated from and what markers was the isolation based on?
 - e. Using the $\beta 1/\beta 2$ -knock out mice, the authors describe a more prominent effect on microglia morphology in the CA region compared to the DG. Why are norepinephrine concentrations then measured in the DG and not in the CA?

Referee #2 (Remarks for Author):

In the present paper by Xu et al. the authors studied the mechanism of environmental enrichment's anti-inflammatory protection of brain microglia against oligomeric A β . The main finding is that enhanced beta-adrenergic signaling is causal for microglial protection. The authors have used a variety of methods to substantiate this claim and all the results speak in the same direction. Oral treatment with isoproterenol (beta-adrenergic receptor agonist) mimicked the beneficial effect on microglia in mice housed under standard condition, while treatment with propranolol (beta-adrenergic receptor antagonist) weakened this protection in mice housed in an enriched environment. Furthermore, beta1/beta2 knock out mice lacked this beneficial effect of environmental enrichment on microglia, hereby emphasizing the important role of beta-adrenergic signaling in this context. The topic is of interest and the study was well conducted and presented. The manuscript is well-written and the conclusions are right to the point, thus the paper seems suited for EMBO Molecular Medicine. The authors could tackle the following comments with a straightforward revision:

- 1) I would recommend shortening the introduction to a maximum of 1 1/2 pages; here, the authors should focus here on the main important information. To the same end, the discussion should be shortened as well.
- 2) On page 6 the authors state that they administered the isoproterenol via the drinking water. Assuming, that several mice are housed together in 1 cage (especially within the context of the environmental enrichment paradigm), the amount of isoproterenol per mouse is difficult to define

and might vary from mouse to mouse. This methodological limitation should at least be mentioned in the text.

3) Concerning Figure 1, Figure 3 and Figure 6: the morphological changes in microglia are very difficult to identify in overview images and need to be displayed in higher magnification- possibly in 3D- with the image analysis software IMARIS.

Minor typos:

- 1) On page 6, in the results headline microglia inflammation in vivo (the space between inflammation and in vivo is missing).
- 2) On page 23, first sentence of the result section: b instead of beta-adrenergic
- 3) On page 30, the last reference is incomplete; volume number and pages are missing.

Referee #3 (Remarks for Author):

Xu et al. hypothesize that environmental enrichment (EE) stimulates endogenous β -adrenergic receptor signaling to modulate microglia function in Alzheimer's disease (AD). They previously demonstrated that EE activates β 2-adrenergic receptor (β 2-AR) signaling in neurons as part of a neuroprotective effect against oligomeric amyloid- β (oA β)-induced toxicity. In turn, oA β induces loss of β 2-ARs in the hippocampus, suggesting links that β 2-AR signaling has with both EE and oA β toxicity. More recently, they demonstrate that EE protects microglia from oA β -induced inflammation. In the current study, they show that the β -AR agonist isoproterenol mimics the anti-inflammatory effect of EE and prevents standard housing (SH) mice from developing microglial inflammation following exposure to oA β . In contrast, the β -AR antagonist propranolol decreases the protective effect of EE, and the mice developed microglial inflammation following exposure to oA β . These data demonstrate that the effect of EE on microglial inflammation is partially mediated by β 1AR and/or β 2AR signaling. Although these studies identify β 1/ β 2ARs as potential modulators of microglial inflammation, these studies are only a minor step forward from their previous publications. A more significant advancement would be to provide mechanistic insight into the involvement of β 1/ β 2AR signaling in regulation of microglial function.

Specific points:

1. The authors indicate changes in microglial morphology, e.g. branching complexity, under different conditions. However, it is also important to determine the effects on microglial density under the conditions described in the manuscript.
2. In the Nanostring study, SH is compared to SH+isoproterenol and EE is compared to EE+propranolol. All conditions were analyzed following oA β treatment and were normalized to whole control group (ID-AD-TBS) for each treatment. A few things are not clear about these experiments:
 - What is the effect of EE vs. SH on the microglial gene profile following oA β treatment?
 - In the absence of oA β treatment, what is the effect of SH vs. EE on the microglial gene expression profile?
 - Does isoproterenol and/or propranolol affect the microglial gene expression in the absence of oA β treatment?
 - Is a similar profile of genes changed following EE alone in comparison to isoproterenol treatment under SH conditions?
3. Do the changes in mRNA levels of the most significantly changed genes correlate with changes in protein levels?
4. Please show protein expression data that microglia express the β 1AR and β 2AR and that oA β induces a decrease in protein levels.
5. To address the mechanism downstream of β -AR activation, is cAMP/PKA activation involved?

6. Does $\alpha\beta$ affect internalization/recycling of the β -ARs following treatment with isoproterenol or propranolol?

1st Revision - authors' response

07 June 2018

Referee #1 (all referees' comments are in italics throughout).

Xu et. al. investigated the role of β -adrenergic signalling via norepinephrine upon soluble Ab - induced microglial activation in the context of environmental enrichment. In this study, the authors aimed to assess the molecular mechanism behind the benefits of physical and mental stimulation resulting in decreased risk of developing Alzheimer's disease (AD). This topic is, per se, of interest to the AD community as it might propose novel therapeutic avenues for the treatment of AD. The authors, in line with previous work published by the group, argue that environmental enrichment (EE) of mice leads to reduced activation of microglia (assessed by microglial morphology and mRNA expression of pro-inflammatory genes) when exposed to soluble oligomeric Ab species extracted from AD brains. The authors demonstrate that this protective effect can be replicated in standard housed (SH) mice by activating β -adrenergic receptors via isoproterenol (Figs. 1, 2). Similarly, blocking β -adrenergic receptors by propranolol administration in EE housed mice prevented the protective effect of EE (Figs. 3, 4). Next, mRNA expression of selected cytokines was replicated by administering a more purified patient-derived soluble oligomeric Ab fraction (Fig. 5). To genetically confirm a role of noradrenergic receptors in this setting, the authors analysed b1/b2-receptor knock out mice kept under SH or EE and treated with soluble Ab; here, the authors show that the beneficial effects of EE were not induced in these mice (Fig. 6). Lastly, the authors provide evidence that the beneficial effects of EE are mediated by the maintenance of norepinephrine signalling that is impaired in SH mice (Fig. 7). The authors thus claim that inducing β -adrenergic receptor activation pharmacologically or via physical or mental activity can prevent the damaging effects of microglia activation by Ab oligomers. As stated above the paper addresses an important topic in the field of AD research. However, there are some major concerns regarding the design of the study.

We thank Referee 1 for nicely summarizing our key findings and highlighting the importance of the topic and its potential impact on developing novel therapeutic avenues for the treatment of AD. Below are our responses to Referee 1's comments.

Major:

1. *In order to draw conclusive data from the experiments, it would be required to have one single AB preparation paradigm and a consistent control for assessing both microglia morphology and gene expression:*
 - a. *In Fig. 1 and 3, microglial morphology is analysed subsequent to i.c.v. injection of soluble protein fractions from control and AD patients (Control-TBS vs AD-TBS).*
 - b. *In Fig. 2 and 4, microglial inflammatory gene expression profiles are assessed following i.c.v. injection of soluble protein fractions from AD patients with or without AB immunodepletion (ADTBS vs ID-ADTBS).*
 - c. *In Fig. 5, microglial gene expression of selected inflammatory genes is analysed after i.c.v. injections of more purified AB preparations (PBS vs SEC-AB).*

At least a thorough characterisation of the employed AB fractions is required. The following questions should be addressed:

- a. *Is the amount of the soluble Ab comparable to those used in the experiments in Fig.1 and 2 and in Fig. 3 and 4, respectively?*
- b. *Does the TBS fraction from the control patients also contain Ab?*

We chose to use control brain TBS extract or immunodepleted ADTBS extracts as two distinct but complementary negative controls, based on a comprehensive consideration of biological relevance and the sensitivity of the experimental methods. Control brain provides the best representation of a healthy brain environment free of AD-type influence (they have no Ab in these extracts by sensitive ELISAs), but person-to-person variations that are independent of AD pathology increase the risk of having microglia-active molecules that are not relevant to AD pathology, especially in highly sensitive, large-scale screens such as Nanostring mRNA profiling. Therefore,

we chose to use control-TBS for morphological studies which approximate the overall biological state of microglia, while using AD-TBS immunodepleted of Ab (ID-ADTBS) for detailed microglial RNA profiling.

Importantly, for Figures 1-4, all the AD-TBS used was prepared using one typical AD human brain. ID-ADTBS used for Figures 2 and 4 was prepared from this same AD-TBS preparation. Control-TBS was prepared from a neuropathologically normal brain. We routinely perform an Abx-42 ELISA developed in our lab (Yang et al., 2013) to characterize all human brain extracts used for experiments. Our ELISA results indicate that while AD-TBS has high amounts of Abx-42, ID-ADTBS only has minimal amount of residual Abx-42 (<4%) and control-TBS has no detectable Abx-42 signal. Therefore, we are confident that the differences in soluble Ab amount between AD-TBS and the two negative control preparations are dependable. We have now added more detailed explanations regarding the two negative controls to the manuscript (Page 7) and included an ELISA graph to Expanded View figure EV1.

As for the SEC-purified oAb for Figure 5, the injection was adjusted by ELISA to precisely 4 pg of Abx-42 per mouse. This injection amount was optimized by a dose study in wild-type SH mice based on their microglial cytokine response, independent of any data acquired with ADTBS. The ELISA readouts from AD-TBS and SEC-purified oAb are not directly comparable because the ELISA capturing antibody has different affinity towards Ab in different forms (monomer, dimer, oligomer, etc.). Therefore, the appropriate injection concentration is determined by biological response rather than biochemical characterization.

c. Are cytokines or cell debris present in the protein fractions?

We prepared AD-TBS by high speed homogenization in Tris buffer saline followed by 50,000g x 1hr ultracentrifugation (Shankar et al., 2008; Yang et al., 2013). All cell debris and non-water-soluble proteins are NOT present in the final supernatant. Control-TBS is prepared identically. Cytokines are present in all such whole tissue saline extracts, but by analyzing all injection materials used in this study with LEGNEDplex™ Human Inflammation Panel for human inflammatory cytokines, we found minimal differences among AD-TBS, control-TBS, and immunodepleted AD-TBS, indicating that human cytokines in the injection material are not responsible for differences in mice inflammatory responses. This is also supported by the prevention of any differences by highly selective Ab immunodepletion. The data are included in the manuscript Appendix (Figure S2). The corresponding description is included in the manuscript (page 7-8). Related method is included on page 24.

d. Why is the contralateral hemisphere used for analyses? Does the injected AB reach into the contralateral side? This needs to be confirmed experimentally prior to assessing those regions, which were used for the analyses of microglial morphology.

As explicitly addressed in our previous publication (Xu et al, *J Neurosci* 2016), the ipsilateral hemisphere understandably experiences local tissue injury from the intraventricular (i.c.v.) injection and could present confounding inflammatory reactions irrelevant to Ab. It has been shown that soluble A β oligomers infused by i.c.v. are fully capable of diffusing in the aqueous CSF and inducing various cellular and biochemical effects in the contralateral hemispheres very shortly after injection. For example, Walsh et al., *Nature* 2002, and Klyubin et al., *Nature Medicine* 2005 show LTP suppression and impaired synaptic plasticity in contralateral rat hippocampus within 1 hr of i.c.v. delivery of oligomeric human Ab (Klyubin et al., 2005; Walsh et al., 2002), demonstrating that diffusible Ab oligomers are directly present in the contralateral brain and cause acute effects, rather than staying local at the injection site. To further address this issue experimentally, we performed unilateral i.c.v. injection of FITC-tagged synthetic Ab42 peptide to visualize its distribution after injection. Brain sections show FITC signal visible within both the ipsilateral and contralateral ventricle as well as the mid-region connecting the two ventricles when visualized 3 hr after injection, with no noticeable difference in signal strength, indicating Ab injected into one lateral ventricle indeed diffuses rapidly to the contralateral side. Therefore, the contralateral hemisphere gets comparable exposure to soluble Ab via the continuous CSF flow between the ventricles and is also free of local tissue injury, providing a superior area for analysis. The data showing FITC-Ab42 presence in both ipsi- and contralateral ventricles is now included in the manuscript as Appendix Figure S1, with corresponding description on page 6.

2. Figs. 2, 4, 5: The relevance of the NanoString data should be discussed in more detail.

a. Why are the five selected genes verified by qPCR most representative for microglial activation and of utmost relevance for AD?

We clarify that by no means do we claim the five selected genes are the *most* representative of microglial activation and of *utmost* relevance for AD in comparison to others. We had briefly stated that we chose the five genes for their large and consistent effect sizes in multiple treatment Ab paradigms and their identity as cytokines implicated in oAb-induced microglial inflammation; they are thus reliable and easily detectable representatives for this and future studies. In more detail, the five genes were selected based on the following criteria: 1) all 5 genes are significantly altered in both our SH isoproterenol vs. water paradigm and our EE propranolol vs. water paradigm, with similar ranking of percentile change; and 2) all genes are major cytokines/chemokines secreted by microglia, and their biological functions are well studied. The main point for selecting the above 5 genes is to create a small, defined set of markers that can be reliably and consistently applied in further assays as representatives to determine the cytokine status of microglia toward oAb. Whether they are more relevant to AD than other highlighted genes with smaller effect sizes (percentile responses) is impossible to say; almost surely, multiple cytokines and other inflammatory molecules are involved in the complex response of microglia to oAb. We have now included some of these detailed explanations in the manuscript (Page 11).

b. Do the differentially expressed genes give any indication on downstream functional modifications?

We ran *all* significantly changed genes through both the NIH DAVID gene functional clustering program and the PANTHER program. Both programs suggested that the genes are quite evenly distributed across multiple major inflammatory pathways, such as IFN-signaling, TNF-signaling, and Toll-like receptor signaling, with the SH water group and EE propranolol group consistently showing more inflammatory activities after oAb injection than do the SH isoproterenol group and the EE water group. This is the central finding of our study. We believe such results indicate that oAb orchestrates a multifaceted immune alteration rather than stimulating a fairly specific pathway in microglia which is neutralized by EE or isoproterenol.

c. A NanoString analysis should be included for SEC-AB treated mice.

We have included the Nanostring analysis as requested in Expanded View Figure EV4, with the corresponding description on page 12.

d. Why are CD68 and P2ry12 not upregulated, which would be expected based on the IHC?

Our imaging data showed an increase of CD68 under more oAb-induced inflammatory conditions. Our quantification methods for microglia morphology do not reflect the expression level of P2ry12, so we cannot claim that P2ry12 expression increased following any treatment. Each available Nanostring code-set has a fixed set of inflammatory genes. CD68 and P2ry12 are not included in the set, so their expression profiles are not part of the dataset. In response to this question, we have now performed qPCR analysis specifically on these two genes and observed CD68 expression patterns in full agreement with the immunostaining results: that the expression levels are increased in the SH control mice and the EE propranolol-treated mice after oAb exposure but not in the SH isoproterenol treated mice or the EE control mice. P2ry12 showed similar trends but was much more variable (see Figure R1 for the reviewers just below; **** $p < 0.00001$). We chose not to present these data in the manuscript because 1) we have already shown their *protein-level* regulation using immunohistochemistry, which we feel is more functionally relevant than mRNA quantification; and 2) including additional genes manually selected into the Nanostring RNA profile would undermine the integrity and unbiased nature of that profiling.

Figure R1. qPCR quantification of microglial CD68 and P2ry12 expression.

e. Additional data on cytokine protein expression is required to support the notion that the treatment induces neuroinflammation, especially since the expression changes are pretty moderate.

In our previous work on this topic (Xu et al, *J Neurosci* 2016), we demonstrated increases in cytokine protein expression of CCL3, CCL4 and TNF α in mouse brain following oAbrich 7PA2 conditioned medium (a CHO cell line secreting soluble Ab peptides) using *in vivo* microdialysis. To better match with the microglia RNA profiling paradigm presented in the current manuscript, we have now performed cytokine ELISAs on brain tissue lysates after i.c.v. injection of ADTBS vs ID-ADTBS. The tissues were harvested following the same procedure as that for Figures 2 and 4. Our data show significant increases by ELISA of CCL2, CCL3, CCL4 and CXCL10 protein expression in the ADTBS injected brains over the ID-ADTBS injected brains. We were not able to get a reliable, above fitting curve reading for TNF α . A colleague's samples harvested from mouse brains after systemic injection of poly I:C, a model for viral infection, showed comparable or higher expression of CCL2, CCL3, CCL4, and CXCL10 than our oAstimulated brain in the exact same ELISA run but also failed to yield any reliable TNF α signal. Thus, the unsuccessful detection of TNF α is a result of its low abundance and the limitation in detection range of that ELISA. We believe our protein level data on the 4 highlighted cytokines are clearly sufficient to confirm the occurrence neuroinflammation and prove that the changes in RNA profile are also reflected in corresponding protein levels. The data are presented in the manuscript Expanded View as Figure EV2, with the corresponding description on page 11. The experimental method is included on page 24.

3. Fig. 5 The authors claim in the text that they "were able to recapitulate the results obtained from the *in vivo* studies of oral isoproterenol or propranolol treatment in SH mice or EE mice". The authors do not show any data supporting this claim; such data must be included.

With due respect, we do not agree with this comment. We have presented qPCR analysis of the highlighted 5 cytokines in Figure 5 (*SEC-purified oAb*). The qPCR result showed highly consistent gene expression patterns for all 5 genes among our 4 housing/treatment paradigms which agree with the Nanostring analysis done using the *whole AD-TBS brain* extract. By "recapitulate" we are aiming at conveying the message that we were able to observe the same gene behavior using both purified and unfractionated human Ab. We apologize if the word "recapitulate" caused confusion. We have now rewritten this part to increase clarity and, as stated above, included the Nanostring profile data (page 12).

4. Fig. 6 The authors argue for a specific effect of norepinephrine/noradrenergic receptor signalling on microglia, yet off-target effects are not assessed.

a. The authors refer to previous work demonstrating the expression of β -adrenergic receptors on microglia, however, these receptors are also expressed on astrocytes or nerve cells. These cells must therefore be analysed as well to see how they are affected by the different treatment paradigms.

We apologize for the confusion about specificity of norepinephrine/noradrenergic receptor signaling on microglia. It was never our intention to claim that the norepinephrine signaling is specific to microglia (we now added the word “global” in the description of genetic modification to avoid future confusion—page 13). In fact, as stated in both Introduction and Discussion, our study is based on the previous findings in our lab (Li et al *Neuron* 2013) that norepinephrine signaling is crucial to neurons in the enriched housing paradigm (Page 4 and Page 22). We agree that by removing b1/b2AR from the mice, multiple cell types will be affected, but our focus for this study is on microglia. Indeed, in our previous work, we performed a comprehensive analysis of neuronal participation of b2AR in the EE housing paradigm (Li et al., 2013). Further, recent work in our lab has shown by electrophysiology that mice with the b2AR KO have normal baseline LTP in standard housing but no longer gain EE benefits on LTP (data unpublished). To address the referee’s question regarding astrocytic responses, we have now performed immunofluorescent microscopy on brain sections used for our microglia analysis in Figure 6, by staining the tissue with an astrocyte marker, GFAP. We performed the same set of morphological analyses on astrocytes as we had on microglia but did *not* identify any significant morphological changes under our paradigm (Figure R2 for the referee, just below).

Figure R2. Morphological analysis on GFAP⁺ astrocytes in Adrb1/2 KO mice in SH and EE environments showed no significant morphological changes after ADTBS i.c.v. injection. The brain sections used for this experiment are from the same mice used in the manuscript for the microglia analysis (Figure 6).

To be more thorough, we further analyzed astrocytes in hippocampus in brain sections from wild-type mice from the SH isoproterenol vs. water and the EE propranolol vs. water paradigms. Again, we did *not* observe any significant morphological differences within each paradigm (Figures R3 and R4, below). Note that our injection and treatment paradigms were designed towards optimal microglial analysis. Our negative data in astrocytes does not suggest that astrocytes do not respond at all to EE, oAb challenge, or adrenergic signaling manipulation. Because the focus of our entire study is microglia, we decide not to include these initially negative astrocyte data in the manuscript for they cannot be viewed as definitive.

Figure R3. Morphological analysis of GFAP⁺ astrocytes in SH mice treated with isoproterenol vs. water showed no significant morphological changes upon AD-TBS i.c.v. injection. The brain sections used for this experiment are from the same mice used in the manuscript for microglia analysis (Figure 1).

Figure R4. Morphological analysis of GFAP⁺ astrocytes in EE mice treated with propranolol vs. water showed no significant morphological changes upon ADTBS injection. The brain sections used for this experiment are from the same mice used in the manuscript for microglia analysis (Figure 3).

b. A microglia-specific b1/b2-receptor knock out would provide a better basis for the role of noradrenergic signalling in microglia and would prove that reduced microglia activation is not just a bystander of modulated noradrenergic signalling. Such -admittedly involved experimental prove is required according to the opinion of this referee.

With due respect, we believe this proposed experiment is simply not feasible both within the provided revision time and with the current technology and understanding of microglia biology. We agree that an ideal mouse model with only microglia b1/b2AR selectively deleted could be of great help in pinpointing b1/b2AR's functions in microglia, however a microglia-specific b1/b2AR knockout mouse has, to our knowledge, never been reported. Current literature has only reported cre-flanked b1AR mice (Mani et al., 2016) and cre-flanked b2AR mice (Jensen et al., 2016) separately. Mice expressing Cre under the CX3CR1 promoter are commercially available. In theory, crossing these 3 lines may create a mouse line with the closest genetic background to a microglia-specific b1/b2AR knockout, as Referee 1 proposed. However, even if such a model existed, it would have multiple caveats. 1) CX3CR1 is not strictly microglia specific. CX3CR1-cre will also be expressed by other myeloid cells such as macrophages in the peripheral system which undercuts the whole point of creating such a line; 2) CX3CR1 itself has important functions in microglial development. Replacing one or both copies of CX3CR1 may lead to changes in microglial functions that add confounding effects which are clearly beyond the scope of our study. Since no gene is yet known to be strictly microglia-specific and can be freely replaced by Cre without influencing microglial development, a pure genetic approach is not available. AAV with microglia-specific serotype is only recently reported, and the authors noted that the efficiency was very low *in vivo* (Rosario et al., 2016). In summary, current technology and knowledge simply cannot provide a mouse line with clear superiority over the germline full knockout that we used. Therefore, a microglia-specific b1/b2AR knockout mouse line would be highly impractical for us to develop within a reasonable timeline. Further, a global germline Adrb1/2KO mouse model fits better with our drug treatments (isoproterenol and propranolol) paradigms as both drugs would target all cell types expressing b1/b2AR. From a translational perspective, we believe the fact that the adrenergic signaling manipulation occurs in cell types other than microglia does not undermine the significance of our finding that EE tempers microglial inflammation via adrenergic signaling.

c. Microglia of the knockout mice seem to have no increased activation compared to the wild type microglia shown in Figs. 1 and 3. What is the author's explanation for this?

First of all, data from the wild-type mice and knockout mice are not directly comparable due to different genetic backgrounds and experiments, so we cannot directly conclude whether the knockout mice have higher or lower activation compared to the wild-type. All our WT mice came from BL6/C129 cross while the b1/b2AR KO mice have a more complex, mixed genetic background. Such differences in genetic background may lead to differences in inflammatory reactions. Further, our previous work has shown that the form of Ab is critical for its biological activity (Yang et al., 2017). It is unreasonable to expect two purified Ab aliquots to contain identical distributions of various lengths of Ab peptides during the reconstitution phase; therefore, the biological effects of two independently conducted experiments are not directly comparable.

On the other hand, we do not expect the microglial activation to be stronger in the knockout mice. There is no definite evidence that endogenous adrenergic signaling actively suppresses inflammation. All the evidence by far, both *in vitro* and *in vivo*, has only pointed to the anti-inflammatory power of *induced* norepinephrine stimulation, either by exogenously administered compounds or EE. Indeed, our data have shown that upon oAb injection, the SH mice suffered a marked loss of extracellular norepinephrine and adrenergic receptors, suggesting a lack of endogenous adrenergic signaling. Therefore we do not think a difference in basal activation intensity should be expected in the knockout mice under our paradigm even if the several caveats we list above did not exist.

Minor points

1. Further editing/proofreading of the text should be undertaken to ensure complete understanding of the messages presented (e.g. first sentence of abstract).

We have carefully edited the entire manuscript, including the abstract, for clearer meaning.

2. Using an acute AD model the authors should discuss that reduced microglial activation translates to any AD-relevant functional or pathological changes.

We have discussed the benefit and implications of acute AD model both *in vivo* and *ex vivo* in several of our previous papers (Li et al., 2013; Shankar et al., 2008; Xu et al., 2016; Yang et al., 2017). Here, we have now included more discussion in this topic to further highlight the benefits and potential impact on AD and on AD research (page 19-20).

3. Since the methods section mainly refers to previously published papers, it is necessary to provide more extensive methodological details in the text or figure legends, e.g.

a. Only in the paragraph describing Fig. 6 does it become clear how old the mice were when exposed to the varying housing conditions.

b. Which regions are stained in Fig. 1 and 3? Using the B1/B2-knock out mice, in Fig. 6, the authors describe a more prominent effect on microglia morphology in the CA region compared to the DG. Why was this regional differentiation not made before?

We have now included more detailed description earlier in the Results section (page 5,6, and 9). All morphological analysis in Figure 1 and Figure 3 were conducted in the dentate gyrus (DG) of the hippocampus. We focused on this region for reasons stated in our previous work (Xu, *J. Neurosci.* 2016) and in the Introduction of this manuscript: that DG is the only region in the wild-type brain that showed increased microglial number per area after EE and had significant microglial morphological changes upon oAb exposure. The CA regions in the wild-type brains have similar morphological changes but no significant increase in microglial density, which the DG does have. When studying the b1/b2-knockout mice, we observed consistent morphological phenotypes in both CA and DG regions. Because such a mouse model has never previously been reported for microglial study, we decided to report both regions instead of focusing on DG in order to be more thorough. To enhance the flow and clarity of the manuscript, we have now moved the CA-related data in Fig. 6 into Expanded View (Figure EV5).

c. What does the readout of %CD68/microglia indicate? CD68 is present in all microglia. Additionally, red, green and blue channels should be split in Fig. 1, 3 and 5 since a green CD68-positive signal is barely visible.

CD68 is a lysosomal marker. The percentage of CD68 within each microglia directly correlates to the level of phagocytosis activity of the microglia. Therefore, even though virtually all microglia have CD68, the increase in CD68 percentage is a strong indicator of increased phagocytic activity connected to inflammatory activation. We only showed merged images in the main figures because of 3 considerations. **1)** At the magnification used in the manuscript, CD68 signals are not prominent in *non-inflammatory* microglia. It is expected that only under inflammatory conditions will CD68 staining be clearly visible as yellow patches inside red microglia (from the red-green colocalization), such as SH water + ADTBS, and EE propranolol + ADTBS. We believe that showing merged images only does not lose important information. **2)** There are non-microglia-associated CD68 signals which may cause confusion or be misleading when presented as single-channel images. **3)** There are limitations of figure size. Per the request of referee, we do now include single-channel images from Figure 1, 3, and 6 in the Appendix as Figure S3, S4, and S5, respectively.

d. For NanoString analysis, which regions were the microglia isolated from and what markers was the isolation based on?

The detailed description regarding cell harvest is included in our previous paper (Xu et al *J. Neurosci.* 2016) and is now incorporated into the Results section. Briefly, for Nanostring analysis, a 3 mm cube of brain tissue from the contralateral hemisphere symmetrical to the ipsilateral injection site was dissected out of each mouse and used for microglia isolation to maximize the detection of inflammatory signals while getting sufficient number of microglia for analysis. The brain tissue corresponds to the same anatomical region that was used for morphological analysis. The microglia were purified based on CD11b⁺/CD45^{Medium}/Ly6C^{High}/Ly6G⁻ by fluorescence-activated cell sorting (FACS) after Percoll gradients. These selection markers remove peripheral monocytes and neutrophils that would not be excluded by Percoll gradient. We have now included above mentioned details in the manuscript (page 8).

e. Using the b1/b2-knock out mice, the authors describe a more prominent effect on microglia morphology in the CA region compared to the DG. Why are norepinephrine concentrations then

measured in the DG and not in the CA?

As stated above, we typically observe more robust EE benefits in the DG of wild-type mice (as regards both changes in microglia morphology and microglia density). oAb-induced morphological changes were observed in the DG of b1/b2-KO mice without any detectable benefits from EE. Therefore, it was reasonable to also sample norepinephrine from the DG of wild-type animals instead of the CA. It is clearly beyond the scope of our study to determine the biological differences between microglia from the CA and DG areas in both wild-type and b1/b2-KO mice.

Referee #2.

In the present paper by Xu et al. the authors studied the mechanism of environmental enrichment's anti-inflammatory protection of brain microglia against oligomeric Aβeta. The main finding is that enhanced beta-adrenergic signaling is causal for microglial protection. The authors have used a variety of methods to substantiate this claim and all the results speak in the same direction. Oral treatment with isoproterenol (beta-adrenergic receptor agonist) mimicked the beneficial effect on microglia in mice housed under standard condition, while treatment with propranolol (beta-adrenergic receptor antagonist) weakened this protection in mice housed in an enriched environment. Furthermore, beta1/beta2 knock out mice lacked this beneficial effect of environmental enrichment on microglia, hereby emphasizing the important role of beta-adrenergic signaling in this context. The topic is of interest and the study was well conducted and presented. The manuscript is well-written and the conclusions are right to the point, thus the paper seems suited for EMBO Molecular Medicine. The authors could tackle the following comments with a straightforward revision:

We thank Referee #2 for his/her positive comments on our work. Our responses to each of the comments are listed below:

1) I would recommend shortening the introduction to a maximum of 1 1/2 pages; here, the authors should focus here on the main important information. To the same end, the discussion should be shortened as well.

We agree and have now edited the manuscript and removed parts that are not directly related to the study.

2) On page 6 the authors state that they administered the isoproterenol via the drinking water. Assuming, that several mice are housed together in 1 cage (especially within the context of the environmental enrichment paradigm), the amount of isoproterenol per mouse is difficult to define and might vary from mouse to mouse. This methodological limitation should at least be mentioned in the text.

We have now incorporated the discussion of this possibility in both the Methods section and Result section (page 6, 9, and 24). If such “competition” for water occurs, it would likely increase inter-animal variability of our results, but nonetheless we were able to observe highly statistically significant effects.

3) Concerning Figure 1, Figure 3 and Figure 6: the morphological changes in microglia are very difficult to identify in overview images and need to be displayed in higher magnification-possibly in 3D-with the image analysis software IMARIS.

We have now performed 3D image reconstruction with IMARIS to highlight two cells from each image at higher resolution and magnification. The IMARIS images are beautiful (we think) and are incorporated into each figure respectively (page 7, 9, 13 and corresponding figure legends). Method description is added to page 25.

Minor typos:

1) On page 6, in the results headline microglia inflammation in vivo (the space between

inflammation and in vivo is missing).

- 2) *On page 23, first sentence of the result section: b instead of beta-adrenergic*
- 3) *On page 30, the last reference is incomplete; volume number and pages are missing.*

We have fixed all the typos listed here.

Referee #3

Xu et al. hypothesize that environmental enrichment (EE) stimulates endogenous b-adrenergic receptor signaling to modulate microglia function in Alzheimer's disease (AD). They previously demonstrated that EE activates b2-adrenergic receptor (b2-AR) signaling in neurons as part of a neuroprotective effect against oligomeric amyloid-b (oAb)-induced toxicity. In turn, oAb induces loss of b2-ARs in the hippocampus, suggesting links that b2-AR signaling has with both EE and oAb toxicity. More recently, they demonstrate that EE protects microglia from oAb-induced inflammation. In the current study, they show that the b-AR agonist isoproterenol mimics the anti-inflammatory effect of EE and prevents standard housing (SH) mice from developing microglial inflammation following exposure to oAb. In contrast, the b-AR antagonist propranolol decreases the protective effect of EE, and the mice developed microglial inflammation following exposure to oAb. These data demonstrate that the effect of EE on microglial inflammation is partially mediated by b1AR and/or b2AR signaling. Although these studies identify b1/b2ARs as potential modulators of microglial inflammation, these studies are only a minor step forward from their previous publications. A more significant advancement would be to provide mechanistic insight into the involvement of b1/b2AR signaling in regulation of microglial function.

We thank Referee #3 for acknowledging our conclusions to date and appreciate the suggestions for further mechanistic studies. Our detailed responses are listed below:

1. The authors indicate changes in microglial morphology, e.g. branching complexity, under different conditions. However, it is also important to determine the effects on microglial density under the conditions described in the manuscript.

We have now performed quantification of microglial density as suggested and the data are now incorporated into Figures 1, 3, and 6 (page 6-7, 9-10, 13 and corresponding figure legends). Briefly, we have observed decreases of microglial density in oAb-exposed SH control mice and in the propranolol treated EE mice. Isoproterenol treated SH mice and control EE mice both showed full rescue of these decreases. In contrast, b1/b2-KO mice showed similar degrees of microglial density decrease in both SH and EE mice. All of these new observations are consistent with our findings in the previous publication (Xu et al., 2016) and this manuscript. We thank Rev. 3 for this helpful suggestion.

2. In the Nanostring study, SH is compared to SH+isoproterenol and EE is compared to EE+propranolol. All conditions were analyzed following oAb treatment and were normalized to whole control group (ID-AD-TBS) for each treatment. A few things are not clear about these experiments:

a. What is the effect of EE vs. SH on the microglial gene profile following oAb treatment?

EE vs. SH microglial gene profile comparison following oAb treatment was the main focus of our previous publication (Xu et al., 2016), in which we demonstrated that EE significantly neutralizes the microglial inflammatory responses of SH microglia induced by i.c.v. injection of oAb. We chose not to repeat such comparisons in this manuscript because it is published already and not the focus of the current study. We now make clear the availability of these earlier data in the Results text (page 5).

b. In the absence of oAb treatment, what is the effect of SH vs. EE on the microglial gene expression profile?

We observed no significant differences between SH and EE on microglial gene profile without oAb treatment. Such conclusion was obtained from comparing SH vs. EE mice receiving negative control preparations i.c.v. in the Nanostring analysis published in our prior work. We have

now added this information to the Results and Discussion sections (page 8 and 11).

c. Does isoproterenol and/or propranolol affect the microglial gene expression in the absence of oAb treatment?

We did not observe any significant gene profile alterations by isoproterenol or propranolol alone without oAb exposure. We have incorporated this point into the Results and Discussion sections (page 8-9, 11, and 18).

d. Is a similar profile of genes changed following EE alone in comparison to isoproterenol treatment under SH conditions?

As stated above, we did not observe any significant changes in the microglial inflammatory gene expression profiles from SH controls following either EE alone or isoproterenol alone. Only when the mice were challenged with oAb did we see similarly significant microglial gene profile changes by EE or isoproterenol in comparison to control SH mice.

3. Do the changes in mRNA levels of the most significantly changed genes correlate with changes in protein levels?

We have now demonstrated by ELISA in control SH mice that 4 of 5 highlighted cytokine genes (Ccl2, Ccl3, Ccl4, and Cxcl10) that are most significantly changed in both the SH isoproterenol vs. water paradigm and the EE propranolol vs. water paradigm have matching cytokine protein level increases in brains injected with ADTBS (please see newly included Expanded View Figure EV2 and description on page 11). TNF α could not be reliably detected by ELISA due to its low abundance in total brain lysates. We are confident in concluding that the changes in mRNA levels of the most significantly altered inflammatory genes are correlated with corresponding changes in their protein levels.

4. Please show protein expression data that microglia express the b1AR and b2AR and that oAb induces a decrease in protein levels.

Antibody availability to b1AR and b2AR has long been a big problem. We recently identified one specific lot of b2AR antibody from Abcam that can show quantifiable bands with adult microglia lysate when using SuperSignalTM West Femto developer and pooling microglia cells isolated from 2 whole brains into one sample (one lane) for a Western blot (~10 ug total protein yield from ~500,000 microglia). Unfortunately, we could not find any b1AR antibody that works with such low total protein input. With this newly identified b2AR antibody, we were able to show that when injected i.c.v. with SEC-purified oAb, SH mice microglia show a strong loss of b2AR signal to a non-detectable level, in comparison to those from negative controls. Further, we were able to show that microglia of EE mice showed minimal change in the b2AR band intensity after oAb exposure. These new data fit well with what we reported at the mRNA level in Fig. 7D. The western blot data are now incorporated into Fig. 7 as panel E, with corresponding description on page 15 and the raw images in Appendix Figure S6.

5. To address the mechanism downstream of b-AR activation, is cAMP/PKA activation involved?

Administering cAMP or PKA inhibitors directly to mouse brains *in vivo* has many confounding effects from cell types other than microglia. To ensure that what we observe is strictly microglial, we instead used primary microglia cultures prepared from P0 mice and performed cAMP/PKA inhibitor treatment in culture. Briefly, the microglia were seeded into 96well plates and treated with SEC-purified human oAb with vs. without isoproterenol. Three different drugs targeting the cAMP/PKA pathway were added individually in addition to the isoproterenol: cAMP-Rp (cAMP inhibitor), KT5720 (PKA inhibitor) or the PKA inhibitor peptide 622 (PKA inhibitor). The microglial inflammatory status was evaluated by qPCR on the 5 key genes highlighted by our earlier Nanostring studies. Among these three drugs, only cAMP-Rp significantly reversed the anti-inflammatory benefits of isoproterenol on Tnf expression but not the other cytokines. Both of the

PKA inhibitors showed no effects on the isoproterenol response. These data indicate that the downstream mechanism of b-AR induced antiinflammatory effects in microglia against oAb is only partially dependent of cAMP and independent of PKA. The data are now incorporated into the main figures as Panel F of Figure 7, with corresponding description on page 15-16 and updated figure legends. Related methods are included on page 26.

6. Does oAb affect internalization/recycling of the b-ARs following treatment with isoproterenol or propanolol?

We tried to perform surface biotinylation on primary microglia cultures to answer this question. However, due to low cell yield from primary microglia cultures (~1x10⁵ viable microglia per litter) and the low antibody sensitivity (see Point #4 above), we were not able to detect any bands by western blot. As mentioned above, we were only able to get a faint quantifiable band with total protein from approximately 5 x 10⁵ microglia. Cellular labeling by the antibody requires detergent; therefore, using immunohistochemistry under detergent-free condition is not an option. We believe these technical limitations make it impossible to answer the question at the moment. Based on our previous study with primary neuron cultures and what we observed here in total microglial protein level changes, we can speculate that oAb causes internalization of b-AR in microglia, leading to decreased available surface receptors. To answer the question of whether isoproterenol enhances the recycling of the b-AR or increased signaling intensity through the available b-AR on the cell surface is beyond currently feasible experimental approaches.

Jensen, C.J., Demol, F., Bauwens, R., Kooijman, R., Massie, A., Villers, A., Ris, L., and De Keyser, J. (2016). Astrocytic beta2 Adrenergic Receptor Gene Deletion Affects Memory in Aged Mice. *PLoS one* 11, e0164721. Klyubin, I., Walsh, D.M., Lemere, C.A., Cullen, W.K., Shankar, G.M., Betts, V., Spooner, E.T., Jiang, L., Anwyl, R., Selkoe, D.J., *et al.* (2005). Amyloid beta protein immunotherapy neutralizes Abeta oligomers that disrupt synaptic plasticity in vivo. *Nature medicine* 11, 556-561. Li, S., Jin, M., Zhang, D., Yang, T., Koeglsperger, T., Fu, H., and Selkoe, D.J. (2013). Environmental novelty activates beta2-adrenergic signaling to prevent the impairment of hippocampal LTP by Abeta oligomers. *Neuron* 77, 929-941. Mani, B.K., Osborne-Lawrence, S., Vijayaraghavan, P., Hepler, C., and Zigman, J.M. (2016). beta1 Adrenergic receptor deficiency in ghrelin-expressing cells causes hypoglycemia in susceptible individuals. *J Clin Invest* 126, 3467-3478. Rosario, A.M., Cruz, P.E., Ceballos-Diaz, C., Strickland, M.R., Siemienski, Z., Pardo, M., Schob, K.L., Li, A., Aslanidi, G.V., Srivastava, A., *et al.* (2016). Microglia-specific targeting by novel capsid-modified AAV6 vectors. *Molecular therapy Methods & clinical development* 3, 16026. Shankar, G.M., Li, S., Mehta, T.H., Garcia-Munoz, A., Shepardson, N.E., Smith, I., Brett, F.M., Farrell, M.A., Rowan, M.J., Lemere, C.A., *et al.* (2008). Amyloid-beta protein dimers isolated directly from Alzheimer's brains impair synaptic plasticity and memory. *Nature medicine* 14, 837-842. Walsh, D.M., Klyubin, I., Fadeeva, J.V., Cullen, W.K., Anwyl, R., Wolfe, M.S., Rowan, M.J., and Selkoe, D.J. (2002). Naturally secreted oligomers of amyloid b protein potently inhibit hippocampal long-term potentiation in vivo. *Nature* 416, 5. Xu, H., Gelyana, E., Rajsombath, M., Yang, T., Li, S., and Selkoe, D. (2016). Environmental Enrichment Potently Prevents Microglia-Mediated Neuroinflammation by Human Amyloid beta-Protein Oligomers. *The Journal of neuroscience : the official journal of the Society for Neuroscience* 36, 9041-9056. Yang, T., Hong, S., O'Malley, T., Sperling, R.A., Walsh, D.M., and Selkoe, D.J. (2013). New ELISAs with high specificity for soluble oligomers of amyloid beta-protein detect natural Abeta oligomers in human brain but not CSF. *Alzheimer's & dementia : the journal of the Alzheimer's Association* 9, 99-112. Yang, T., Li, S., Xu, H., Walsh, D.M., and Selkoe, D.J. (2017). Large Soluble Oligomers of Amyloid beta-Protein from Alzheimer Brain Are Far Less Neuroactive Than the Smaller Oligomers to Which They Dissociate. *The Journal of neuroscience : the official journal of the Society for Neuroscience* 37, 152-163.

2nd Editorial Decision

03 July 2018

Thank you for the submission of your revised manuscript to EMBO Molecular Medicine. We have now received the enclosed reports from the referees that were asked to re-assess it. As you will see the reviewers are now globally supportive and I am pleased to inform you that we will be able to accept your manuscript pending the following final amendments:

- 1) Please address the minor comments of referees 1 and 3.

Please provide a letter INCLUDING my comments and the reviewer's reports and your detailed responses to their comments (as Word file).

Please submit your revised manuscript within two weeks. I look forward to seeing a revised form of your manuscript as soon as possible.

***** Reviewer's comments *****

Referee #1 (Remarks for Author):

Xu et. al. investigated the role of β -adrenergic signalling via norepinephrine upon soluble A β -induced microglial activation in the context of environmental enrichment. In this study, the authors aimed to assess the molecular mechanism behind the benefits of physical and mental stimulation resulting in decreased risk of developing Alzheimer's disease (AD) and conclude that inducing β -adrenergic receptor activation pharmacologically or via physical or mental activity can prevent the damaging effects of microglia activation by A β oligomers. This topic is, per se, of interest to the AD community as it might propose novel therapeutic avenues for the treatment of AD.

The authors addressed our comments by adding experiments or rewriting the manuscript. However, minor changes should still be included.

To facilitate the review process the figures should be labelled for identification. It is not clear which figures at the end of the main manuscript belong to which figure legend. Especially because the legends of figures EV1-EV5 are in a separate document.

Major points

1 c Data in Supplementary Fig. S2 show amounts of cytokines between AD-TBS vs ID-ADTBS preparations. Interestingly, IL18 and TNF α in the control TBS are quite high. While this is less than ideal it does not seem to affect microglia activation. Typo: The name of the Human Inflammation Panel should be changed to LEGENDplex in the text.

2 c In Fig. EV4 gene names are missing and must be included.

4 a and b Effects on astrocytes were analysed and effects on neurons are present (published and unpublished data) but differential effects of a knockout of β -adrenergic receptors specific to neurons or microglia were not assessed. The reviewer would like to comment that the authors correctly criticize the CX3CR1-Cre line, yet the caveat of targeting peripheral myeloid cells can either be circumvented by using an inducible CX3CR1-CreERT2 line (Yona et al., 2013, Immunity) or by using the Sal1-Cre line which is microglia-specific (Buttgereit et al., 2016, Nat Immunol). Additionally, using these lines would not result in replacing CX3CR1/Sal1 genes and thus modifying microglia functions, it would only drive Cre expression under these specific promoters. Next to targeting β -adrenergic signaling in vivo, a knockout could be achieved using in vitro methods. There exist other alternatives to AAV-based targeting, such as RNA interference or CRISPR. Whilst the reviewer understands the time restraints in performing such experiments, the authors should discuss this in the manuscript for further possible experiments.

Minor point

3 c The definition of percentage CD68 positive cells as described by the authors should be included in the results section and/or figure legend of Fig. 1.

Referee #2 (Remarks for Author):

The Authors have adequately addressed all the comments that have been raised in a previous round of review so that the manuscript is now acceptable for publication.

Referee #3 (Remarks for Author):

In the revised manuscript, Xu et al. have addressed many of the issues raised by this reviewer from the original version. However, there is still one concern that should be addressed, which would make the manuscript suitable for publication in EMBO Molecular Medicine.

In Fig. 7F, it is unclear why the authors evaluated the mRNA expression levels of the primary cytokines rather than protein levels following treatment with the cAMP and PKA inhibitors. As they have shown by ELISA that $\alpha\beta$ affects the levels of CCL2, CCL3, CCL4, and CXCL10 in Fig. EV2, ELISA analysis of cytokine levels would be more appropriate for this assay.

2nd Revision - authors' response

12 July 2018

Referee #1 (Remarks for Author):

Xu et. al. investigated the role of β -adrenergic signalling via norepinephrine upon soluble $A\beta$ -induced microglial activation in the context of environmental enrichment. In this study, the authors aimed to assess the molecular mechanism behind the benefits of physical and mental stimulation resulting in decreased risk of developing Alzheimer's disease (AD) and conclude that inducing β -adrenergic receptor activation pharmacologically or via physical or mental activity can prevent the damaging effects of microglia activation by $A\beta$ oligomers. This topic is, per se, of interest to the AD community as it might propose novel therapeutic avenues for the treatment of AD.

The authors addressed our comments by adding experiments or rewriting the manuscript. However, minor changes should still be included.

To facilitate the review process the figures should be labelled for identification. It is not clear which figures at the end of the main manuscript belong to which figure legend. Especially because the legends of figures EV1-EV5 are in a separate document.

We have fixed this throughout.

Major points

1. Data in Supplementary Fig. S2 show amounts of cytokines between AD-TBS vs ID-ADTBS preparations. Interestingly, IL18 and TNF α in the control TBS are quite high. While this is less than ideal it does not seem to affect microglia activation. Typo: The name of the Human Inflammation Panel should be changed to LEGENDplex in the text.

We agree that the human cytokines in the brain extracts do not lead to any microglial activation in mice. Endogenous cytokine levels of different human (AD and control) brains can vary greatly and have no correlation with their $A\beta$ load. The fact that human cytokines clearly did not affect mouse microglial morphology in our paradigm (as exemplified by control TBS with high human TNF α and IL18 not inducing any inflammatory morphological changes) means that the morphological changes we observed are strictly due to $A\beta$ stimulation rather than to microglial responses towards cytokines. The typo has been fixed in the article.

2. In Fig. EV4 gene names are missing and must be included.

We have added the gene names to the figure. Our previous concern was that due to the restriction in figure size the gene names won't be visible in the printed version, so we chose not to include the names in the figure but instead provide the full gene list with values in the Expanded View. But we now include the gene names in the figure. The names are still too small to be read in the printed size, but they are clear when the figure is magnified electronically, as readers will be able to do.

3. Effects on astrocytes were analysed and effects on neurons are present (published and unpublished data) but differential effects of a knockout of β -adrenergic receptors specific to neurons or microglia were not assessed. The reviewer would like to comment that the authors correctly criticize the CX3CR1-Cre line, yet the caveat of targeting peripheral myeloid cells can either be circumvented by using an inducible CX3CR1-CreERT2 line (Yona et al., 2013, Immunity) or by using the Sal1-Cre line which is microglia-specific (Buttgereit et al., 2016, Nat Immunol). Additionally, using these lines would not result in replacing CX3CR1/Sal1 genes and thus modifying microglia functions, it would only drive Cre expression under these specific promoters. Next to targeting β -adrenergic signaling in vivo, a knockout could be achieved using in vitro

methods. There exist other alternatives to AAV-based targeting, such as RNA interference or CRISPR. Whilst the reviewer understands the time restraints in performing such experiments, the authors should discuss this in the manuscript for further possible experiments.

We appreciate Referee 1's thoughtful comments. We have now included new discussion about the future possibility of attempting cell-type specific targeting using mice with controlled microglia-specific down-regulation of β -adrenergic receptors (see page 18, lines 13-18).

Minor point

The definition of percentage CD68 positive cells as described by the authors should be included in the results section and/or figure legend of Fig. 1.

We have included the definition of %CD68 in the first part of the Results section where we describe the results presented in Figure 1 (page 6 last line to page7).

Referee #2 (Remarks for Author):

The Authors have adequately addressed all the comments that have been raised in a previous round of review so that the manuscript is now acceptable for publication.

We greatly appreciate Referee 2's support.

Referee #3 (Remarks for Author):

In the revised manuscript, Xu et al. have addressed many of the issues raised by this reviewer from the original version. However, there is still one concern that should be addressed, which would make the manuscript suitable for publication in EMBO Molecular Medicine.

In Fig. 7F, it is unclear why the authors evaluated the mRNA expression levels of the primary cytokines rather than protein levels following treatment with the cAMP and PKA inhibitors. As they have shown by ELISA that αAI^2 affects the levels of CCL2, CCL3, CCL4, and CXCL10 in Fig. EV2, ELISA analysis of cytokine levels would be more appropriate for this assay.

We measured cytokine mRNA levels rather than protein levels because the cell and therefore protein yields from each treatment condition (~20,000 cells/well in 150 μl of culture medium) was simply not sufficient for consistent ELISA measurements of the relevant cytokines we studied. We have now included this explanation in the Results section corresponding to Figure 7F (page 16, line 3-7).

Corresponding Author Name: Dennis Selkoe
Journal Submitted to: EMBO Molecular Medicine
Manuscript Number: EMM-2018-08931